



# Estimation of subsurface soil moisture from surface soil moisture in cold mountainous areas

Jie Tian[1,2], Zhibo Han[1], Heye Reemt Bogena[2], Johan Alexander Huisman[2], Carsten Montzka[2], Baoqing Zhang[1*], Chansheng He[1,3*]

[1]Key Laboratory of Western China's Environmental Systems (Ministry of Education), College of Earth and Environmental Sciences, Lanzhou University, Lanzhou, Gansu 730000, China

[2]Agrosphere Institute (IBG-3), Forschungszentrum Jülich, 52425 Jülich, Germany

[3]Department of Geography, Western Michigan University, Kalamazoo, MI 49008, USA

*Correspondence to*: Baoqing Zhang (baoqzhang@lzu.edu.cn) and Chansheng He (he@wmich.edu)

**Abstract.** Profile soil moisture (SM) in mountainous areas are significant in water resources management and ecohydrological studies of downstream arid watersheds. Satellite products are useful in providing spatially distributed SM information, but only have limited penetration depth (e.g. top 5 cm). In contrast, in situ observations can provide multi-depth measurements, but only with limited spatial coverage. Spatially continuous estimates of subsurface SM can be obtained from surface observations using statistical methods, but this requires sufficient coupling strength among surface and subsurface SM. This

study evaluates methods to calculate subsurface SM from surface SM and an application to the satellite SM product based on a SM observation network in the Qilian Mountains (China) established since 2013. First, we used cross-correlation to analyze the coupling strength among surface (0-10 cm) and subsurface (10-20, 20-30, 30-50, 50-70 cm, and profile of 0-70 cm) SM. Our results indicated an overall strong coupling among surface and subsurface SM in this study area. Afterwards, three different methods were tested to estimate subsurface SM from in-situ surface SM: the exponential filter (ExpF), artificial neural

networks (ANN) and cumulative distribution function matching (CDF) methods. The results showed that both ANN and ExpF methods were able to provide accurate estimates of subsurface soil moisture at 10-20 cm, 20-30 cm, and for the profile of 0-70 cm using surface (0-10 cm) soil moisture only. Specifically, the ANN method had the lowest estimation error (RSR) of 0.42, 0.62 and 0.49 for depths of 15 and 25 cm and profile SM, respectively, while the ExpF method best captured the temporal variation of subsurface soil moisture. Furthermore, it could be shown that the performance of the profile SM estimation was

not significantly lower with using an area-generalized $T_{opt}$ (optimum T) compared to the station-specific $T_{opt}$. In a final step, the ExpF method was applied to the satellite SM product (Soil Moisture Active Passive Level 3: SMAP_L3) to estimate profile SM, and the resulting profile SM was compared to in situ observations. The results showed that the ExpF method was able to estimate profile SM from SMAP_L3 surface products with reasonable accuracy (median R of 0.718). It was also found that the combination of ExpF method and SMAP_L3 surface product can significantly improve the estimation of profile SM in the

mountainous areas in comparison to the SMAP_L4 root zone product. Overall, it was concluded that the ExpF method is able to estimate profile SM using SMAP surface products in the Qilian Mountains.



## 1. Introduction

Soil moisture (SM) is considered an essential climate variable (Bojinski et al., 2014) because of its critical role in the water, energy (Jung et al., 2010) and carbon cycle (Green et al., 2019). In particular, knowledge of profile SM is important for runoff
modeling (Brocca et al., 2010) and water resources management (Gao et al., 2018), drought assessment (Jakobi et al., 2018), and climate analysis (Seneviratne et al., 2010). Methods for SM measurements included ground-based measurements and satellite-based methods (Dobriyal et al., 2012). Most ground-based methods enable the determination of SM changes with high temporal resolution at different depths, but with limited spatial coverage (Jonard et al., 2018). Especially in mountainous regions, in situ SM measurements at large scales are difficult to obtain and thus scarce (Ochsner et al., 2013). In addition,
strong SM heterogeneity in complex mountainous makes SM estimation at large scales more difficult (Williams et al., 2009). By comparison, satellite estimates of SM, such as the Soil Moisture Active & Passive (SMAP), provide spatial SM coverage at large scale (Entekhabi et al., 2014; Brocca et al., 2017). Unfortunately, SMAP and other microwave-based SM products from spaceborne sensors only provide SM estimates for a limited depth up to ~5 cm (Escorihuela et al., 2010). Thus, a gap exists with respect to the availability of deeper SM information with adequate spatial coverage.

Previous studies have shown that deeper SM is often related to surface SM using cross-correlation analysis (Mahmood and Hubbard, 2007; Mahmood et al., 2012; Wang et al., 2017). However, correlation strength depends on location and climate (Ford et al., 2014; Carranza et al., 2018). Mahmood and Hubbard (2007) found that the correlation coefficient between surface (0.5 cm) and subsurface SM from 0 to 60 cm depth ranged from 0.21 to 0.69 for different locations and land covers. Carranza et al. (2018) reported that the coupling strength also depends strongly on SM itself.

A variety of approaches for estimating deep SM from surface SM information have been developed, including data assimilation of remote sensing data into land surface model (Han et al., 2013), physically-based methods (Manfreda et al., 2014), (semi-) empirical approaches (Albergel et al., 2008), data-driven methods (Gao et al., 2013; Kornelsen and Coulibaly, 2014; Zhang et al., 2017a), and statistical methods (Gao et al., 2019). Among them, the application of both data assimilation and physically based methods are limited due to the large amount of required input data, e.g. soil properties, which are often
not available for data-scarce mountainous areas (Jin et al., 2015; Li et al., 2017; Dai et al., 2019). Cumulative Distribution Function (CDF) matching approach is a statistical method that adjusting systematic differences in different SM datasets (e.g. in-situ observations and satellite products) based on observation operators (Drusch et al., 2005; Peng et al., 2017). The CDF matching can be used for upscaling of SM (Han et al., 2012) and estimating subsurface SM based on surface SM recently (Gao et al., 2019). Artificial neural networks (ANN) are effective and powerful data-driven tools for nonlinear estimation problems,
and have been widely used to estimate subsurface SM from surface SM measurements (Kornelsen and Coulibaly, 2014; Pan et al., 2017). The exponential filter (ExpF) method belongs to semi-empirical model on the basis of two-layer SM balance equation (Wagner et al., 1999). This method has been widely applied with both in situ observations and satellite products, and the performance of ExpF method in estimating subsurface SM varied considerably over regions with different environment conditions (Ford et al., 2014; González-Zamora et al., 2016; Tobin et al., 2017; Wang et al., 2017; Zhang et al., 2017a). Ford



et al. (2014) found that the root zone SM estimated from SMOS satellite products had a mean $R^2$ of 0.57 (ranging from 0.00 to 0.86) and 0.24 (ranging from 0.00 to 0.51) for SM networks in Oklahoma and Nebraska, respectively. In addition to surface SM data, the ExpF method requires only one additional parameter (T, the characteristics time length) that reflects the joined influence of local conditions on the temporal characteristics of SM (Albergel et al., 2008; Ceballos et al., 2005). Previous researches have shown that T varied among different stations, and the control of environment factors on T is not conclusive

(Albergel et al., 2008; Lange et al., 2008; Paulik et al., 2014; Wang et al., 2017). Only soil depth has been shown to affect T significantly (Albergel et al., 2008). Despite these findings, methods for estimating deeper SM from surface SM have not been evaluated for high and cold mountainous areas using large scale in-situ SM observations. In the absence of large scale networks of in situ SM observations in mountainous areas, satellite SM products can be an alternative for providing large scale surface SM information (Ochsner, et al., 2013). Although SM estimation from spaceborne sensors is especially challenging for

mountainous regions, some validation activities have shown adequate accuracy (Pasolli et a., 2011; Rasmy et al., 2011; Zhao et al., 2014; Zeng et al., 2015; Zhao and Li, 2015; Colliander et al., 2017; Ullah et al., 2018; Qu et al., 2019; Liu et al., 2019). Nevertheless, the accuracy of profile SM estimation from remotely sensed SM products is currently unknown for mountainous regions.

In this study, we focus on the Qilian Mountains, which is a water source for several key inland rivers with terminal lakes in

Northwest China, including the Heihe, Shiyang, and Shule Rivers (He et al., 2018). Water scarcity threatens both food and ecosystem security in these endorheic basins (Feng et al., 2019). At the northeastern border of the Tibet-Qinghai plateau with its significant role in the Asian monsoon, profile water content in the Qilian Mountains is a key variable in ecohydrological studies on water resources and exchange processes in these basins (Zhao et al., 2013). Therefore, the aim of this study is to use multi-station in situ SM observations and remotely sensed SM data from the Qilian Mountains, which is a prime example of a

high and cold mountainous area, to characterize the relationship between surface SM and deeper SM in order to obtain the spatial distribution of profile SM. We first evaluated the performance of the different methods for estimating subsurface SM. Subsequently, the best method was employed with SMAP surface SM products to evaluate its utility for estimating profile SM for mountainous regions.

## 2. Study Area

This study was carried out in the upland area of the Heihe River Basin, which is a typical terminal lake basin of arid regions (Liu et al., 2018) (Fig 1). It located in the Qilian Mountains at the Northeastern of the Qinghai-Tibet plateau. It covers approximately $2.7 \times 10^4$ km$^2$ and the elevation ranges from about 2000 to 5000 m (Yao et al., 2017). The region has an annual precipitation ranging from 200 to 500 mm (Luo et al., 2016), annual potential evapotranspiration ranges from 700 to 2000 mm, and the annual mean temperature ranged from -3.1 °C to 3.6 °C during 1960-2012 (He et al, 2018). The main land covers are

grassland, forestland and sparse vegetated land (Zhou et al., 2016). The main soil types are Calcic Chernozems, Kastanozems, and Gelic Regosols, and the main soil texture classifications are silt loam, silt and sandy loam (Tian et al., 2017; 2019).

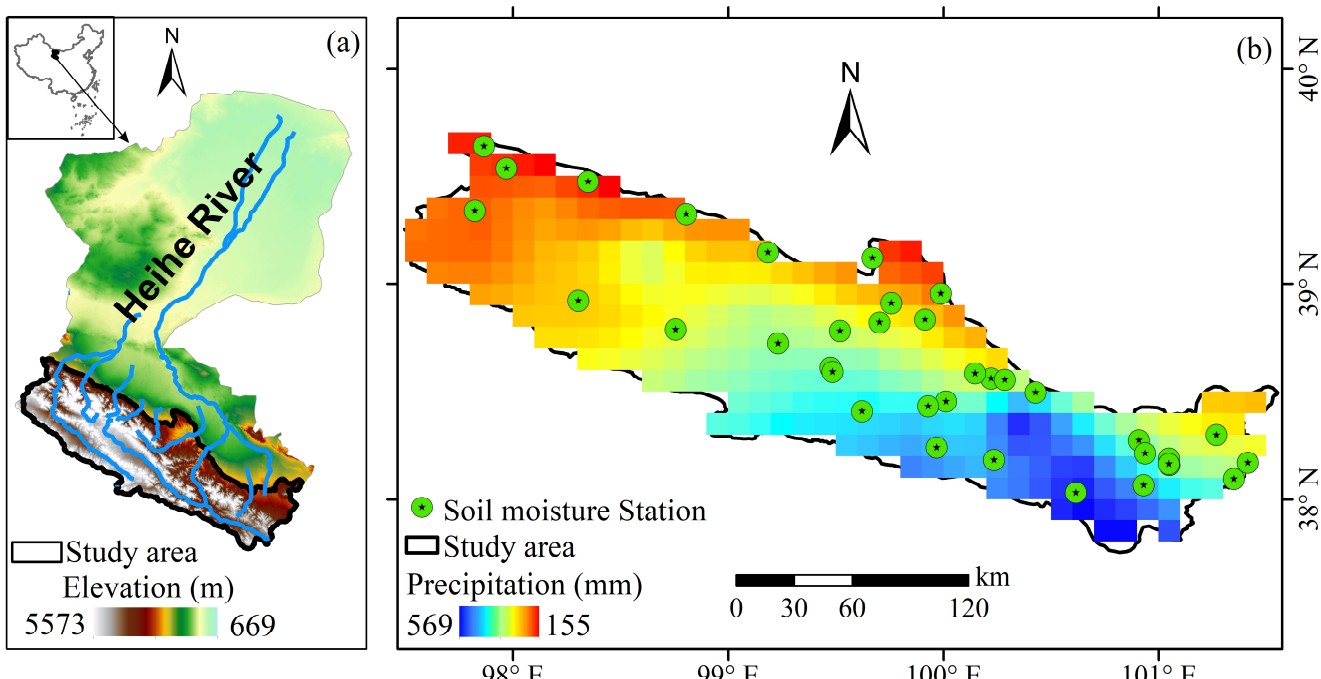

**Fig. 1. (a)** Study area and **(b)** distribution of the SM stations with spatial distribution of annual average precipitation from 2014 to 2016.

## 3. Data and Methods

### 3.1. Datasets

A SM monitoring network was set up in September 2013 in the Qilian Mountains. The network is composed of 35 SM stations distributed over the entire study area (Fig. 1). At each station, SM profiles from 0 to 70 cm were measured by soil moisture probes (ECH2O 5TE, METER Group Inc., USA) installed at depths of 5 (representing depth of 0-10 cm, $SM_{5\ cm}$), 15 (10-20 cm, $SM_{15\ cm}$), 25 (20-30 cm, $SM_{25\ cm}$), 40 (30-50 cm, $SM_{40\ cm}$) and 60 cm (50-70 cm, $SM_{60\ cm}$) below the soil surface at 30 min intervals. Soil-specific sensor calibrations were performed with the direct calibration method (Cobos and Chambers, 2010; Zhang et al., 2017b) using soil samples taken from each station. The profile integrated SM ($SM_{0-70\ cm}$) was calculated by the method of (González-Zamora et al., 2016):

$$SM_{0-70\ cm} = \frac{SM_{5\ cm} \times 10 + SM_{15\ cm} \times 10 + SM_{25\ cm} \times 10 + SM_{40\ cm} \times 20 + SM_{60\ cm} \times 20}{70} \tag{1}$$

The entire data set used in this study thus consists of six in situ SM time series at depths of 5, 15, 25, 40, 60 cm, and 0-70 cm for each of the 35 stations. Due to the influence of soil freezing in winter, the soil moisture time series were limited to the growing season (May to October, Tian et al., 2019) of 2014 to 2016. The half-hourly measurements were averaged to obtain daily SM values that will be used for the estimation of subsurface SM (Wagner et al., 1999). Data quality management was performed for each station, and data gaps existed in the harsh mountainous environment, as described in detailed in Tian et al.





(2019). Time series where the amount of missing values exceeded 50% were excluded in the analysis. The final dataset after
processing is presented in Fig. 2. The surface SM measured at 5 cm was used to predict the subsurface SM at depths of 15, 25,
40, 60 cm and the profile average (0-70 cm).

Soil cores were taken to measure soil properties including soil organic carbon (SOC), saturated hydraulic conductivity ($K_S$),
soil particle composition and bulk density and for each layer during the sensor installation. Detailed descriptions of the soil
properties can be found in Tian et al. (2017; 2019). The statistics of the soil physical characteristics are provided in Table 1.

Daily precipitation from reanalysis (Chen et al., 2011), a 5-day composite of Leaf Area Index (LAI) and Normalized
Difference Vegetation Index (NDVI) data at a resolution of 1 km (Yin et al., 2015) during 2014-2016 were acquired from the
National Tibetan Plateau Data Centre (https://data.tpdc.ac.cn/en/).

The widely used higher level SMAP_L3 Global Daily 9 km product for the growing seasons of 2015 and 2016 were used
in this study (O'Neill et al., 2018), the product has been distributed by NASA (http://nsidc.org/). SMAP descending node
observations acquired near 6:00 AM local solar time have been combined to global daily composites in order to reduce the
impacts of Faraday rotation and to consider the assumption of uniform vertical profiles of soil temperature and soil dielectric
properties. It has to be noted that the data have been posted on the 9 km grid, but it's the result of the Backus-Gilbert optimal
interpolation at brightness temperature level. The real spatial resolution is coarser (O'Neill et al., 2018).

SMAP_L4 contains estimates of both the surface and root zone SM product based on the assimilation of brightness
temperature into the NASA land-surface model (Reichle et al., 2017). The SMAP L4 product has a spatial and temporal
resolution of 9 km and 3 h, respectively, and the products were daily averaged.

**3.2. Cross-correlation analysis**

Cross-correlation method was conducted to investigate time-lag relationships between surface and subsurface SM at different
depths (Georgakakos et al., 1995; Mahmood and Hubbard, 2007). In particular, cross-correlation method was applied to
characterize the coupling among surface SM (5 cm) and subsurface SM (15 cm, 25 cm, 40 cm, 60 cm, and profile of 0-70 cm)
with a maximum lag of 68 days. The maximum lagged cross-correlation coefficients (Lag-R) between two time series and the
corresponding lag time (day) were calculated. A small Lag-R indicates a weak coupling between surface and subsurface SM,
which implies that surface SM has low predictive power for subsurface SM, and vice versa (Mahmood and Hubbard, 2007).
After examining the coupling strength between surface and subsurface SM, three different methods were evaluated to estimate
subsurface SM from surface observations. These three methods are described in detail in the following.



**Fig. 2**. Daily soil moisture (vol. %) time series during the growing season of 2014 to 2016 for the 5 layers (layer 1, 0-10 cm; layer 2, 10-20 cm; layer 3, 20-30 cm; layer 4, 30-50 cm; layer 5, 50-70 cm) in the 35 soil moisture stations. Gaps exist for some stations due to missing data.






**Table 1.** Statistics of the soil physical characteristics at the 35 soil moisture stations: mean (standard deviation)

| Layer | Depth (cm) | Bulk Density (g/cm³) | $K_S$ (cm/hour) | SOC (g/100g) | Sand (%) | Silt (%) | Clay (%) |
|---|---|---|---|---|---|---|---|
| Layer 1 | 0-10 | 1.13(0.28) | 3.87(4.11) | 4.35(4.11) | 26.6(11.88) | 66.19(10.93) | 7.21(1.64) |
| Layer 2 | 10-20 | 1.14(0.24) | 4.61(4.53) | 3.9(3.87) | 24.52(11.93) | 68.57(11.22) | 6.88(1.21) |
| Layer 3 | 20-30 | 1.18(0.32) | 4.78(6.22) | 3.63(3.54) | 27.03(15.21) | 66.48(14.28) | 6.46(1.38) |
| Layer 4 | 30-50 | 1.29(0.3) | 3.94(4.68) | 2.21(2.28) | 29.51(15.29) | 63.81(14.47) | 6.5(1.55) |
| Layer 5 | 50-70 | 1.34(0.3) | 1.85(2.35) | 2.34(2.47) | 26.89(17.06) | 66.54(15.86) | 6.71(1.89) |

Note: $K_S$ is the Saturated Hydraulic Conductivity; SOC is the Soil Organic Carbon.

### 3.3. Exponential Filter (ExpF) method

The ExpF method predicts the dynamics of subsurface SM using exponential filter function of the surface SM dynamics
(Wagner et al., 1999; Albergel et al., 2008). First, SM (cm³/cm³) is transformed into a soil water index (SWI) with:

$$SWI_i = \frac{\theta_i - \theta_{i,min}}{\theta_{i,max} - \theta_{i,min}} \tag{2}$$

where $\theta_{i,min}$ and $\theta_{i,max}$ are the minimum and maximum of SM time series collected since installation of each layer of
each station (Ford et al., 2014). The ExpF method then estimates subsurface SM from surface SM using:

$$SWI_{m,t_n} = SWI_{m,t_{n-1}} + K_{t_n}(ms_{t_n} - SWI_{m,t_{n-1}}) \tag{3}$$

where $SWI_{m,t_{n-1}}$ and $SWI_{m,t_n}$ are the predicted subsurface SWI at time $t_{n-1}$ and $t_n$, respectively. $ms_{t_n}$ is the observed
surface SWI at time $t_n$, and $K_{t_n}$ represents the gain at time $t_n$ calculated by:

$$K_{t_n} = \frac{K_{t_{n-1}}}{K_{t_{n-1}} + e^{-\frac{t_n - t_{n-1}}{T}}} \tag{4}$$

where $K_{t_{n-1}}$ is the gain at time $t_{n-1}$ and T is the characteristic time length in day. The equation was initialized
with $SWI_{m,t_1} = ms_{t_1}$, and $K_{t_1} = 1$ (Albergel et al., 2008). This method is particularly useful as T is the only unknown
parameter. The optimum T ($T_{opt}$) was determined by optimization using the highest Nash-Sutcliffe score for each specific depth
at each station.

### 3.4. Artificial Neural Network (ANN) method

ANN is a data-driven method to predict subsurface SM from surface SM (Zhang et al., 2017a). If properly trained, ANN are
able to describe the nonlinear relationships between the dynamics of SM at different depths (Kornelsen and Coulibaly, 2014).
The commonly used feed-forward ANN (with one hidden layer and 10 neurons, Levenberg–Marquardt algorithm, Ford et al.,
2014) was used in this study., The ANN modelling was carried out using MATLAB (neural network time series tool, R2017b,





The MathWorks). As suggested by Zhang et al. (2017a), 70% of data were randomly selected for training of the ANN and the remaining 30% were used for validation.

### 3.5. Cumulative Distribution Function matching (CDF) method

The CDF method was used for bias correction and the scaling issue of different datasets of measurements (Reichle and Koster, 2004; De Lannoy et al., 2007). It has also been successfully applied for estimating subsurface SM from surface measurements recently (Gao et al., 2019). In this study, the following procedure for CDF matching was used:

1) Rank the surface ($\theta_1$) and the subsurface SM ($\theta_2$) time series;

2) Calculate the difference between the two observation time series:

$$\Delta_i = \theta_{1,i} - \theta_{2,i} \tag{5}$$

3) Use a cubic polynomial fit to relate the difference ($\Delta$) to surface SM ($\theta_1$) as recommended by Gao et al. (2018):

$$SWI_{m,t_n} = SWI_{m,t_{n-1}} + K_{t_n}(ms_{t_n} - SWI_{m,t_{n-1}}) \tag{6}$$

4) Calculate CDF-matched subsurface SM ($\theta_{CDF}$) with:

$$\theta_{CDF} = \theta_1 - \hat{\Delta} \tag{7}$$

Similar to the ANN method, 70% of the data were used to calibrate the approach and the remaining 30% of the data were used for validation of the CDF matching method.

### 3.6. Statistical analysis

Boxplots were used to show the scattering of the data. The difference between data in different groups was examined through the one-way analysis of variance (ANOVA) with the post-hoc Bonferroni test when the normality and homogeneity of variance

of the datasets were satisfied. The Kruskal-Wallis ANOVA with a post-hoc Dunn's test was used in case these conditions were not satisfied (Lange et al., 2008). The statistics were implemented through SPSS (SPSS 18.0, SPSS Inc.) and Matlab (R2017b, The MathWorks). The significance level was 0.05 for all statistical analysis.

## 4. Results and discussion

### 4.1. Cross-correlation analysis

The boxplot of cross-correlation results (Lag-R and lag time) between surface and subsurface SM of all stations is shown in Fig.3. It can be seen that Lag-R decreased with depth and that the associated lag time increased. In particular, Lag-R decreased from a median value of 0.91 for layer 2 with a lag of 0.5 day to a median value of 0.57 for layer 5 with a lag of 7 days. The median value of Lag-R for profile SM from 0 to 70 cm was 0.84 with no time lag. Statistical tests showed that the Lag-R of



layer 2, layer 3 and the profile SM are significantly larger than that of layers 4 and 5. The difference in Lag-R between profile

SM and layer 2 (p=0.4) and layer 3 (p=0.09) were not significant. The observed decrease of Lag-R and increase of lag time

with depth are persistent with preceding researches (Mahmood et al., 2012; Zhang et al., 2017a), and indicates that the coupling

strength between surface and subsurface SM decreased along soil profile (Wu et al., 2002; Carranza et al., 2018). The coupling

strength in our study area (mean value of 0.77 and 0.56 for 20-30 cm and 50-70 cm, respectively) was similar to that of Ford

et al. (2014) (mean value of 0.78 and 0.61 for 25 cm and 60 cm, respectively). These results suggest that there is a strong

coupling strength between surface and subsurface SM in the top 70 cm of soil in our study area.

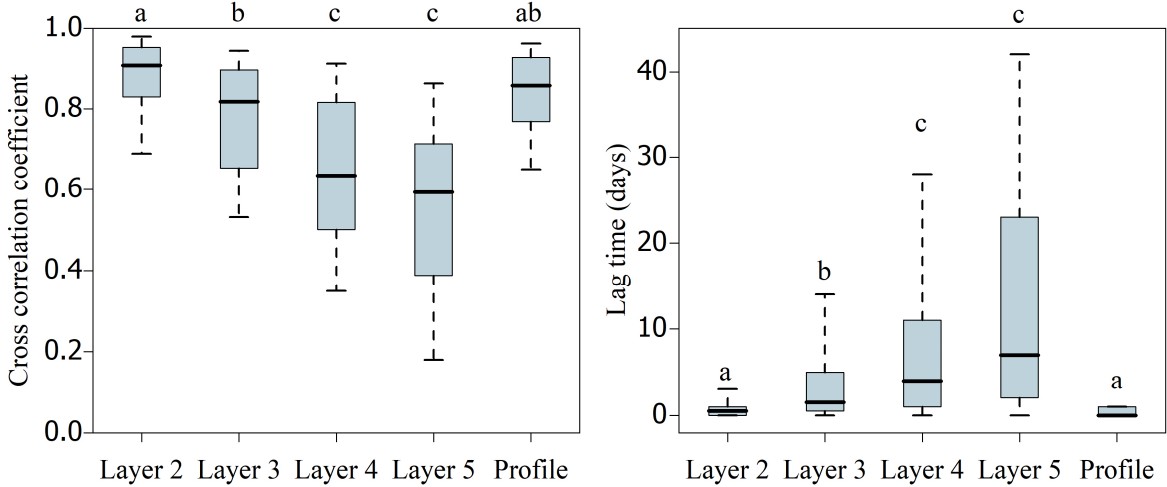

**Fig. 3.** The boxplot of cross correlation coefficient and lag time of different layers for all stations. Different letters mean significant differences between different layers (p<0.05, Kruskal-Wallis ANOVA, post-hoc Dunn's test). The upper (lower) whisker, upper (lower) boundary of box and the line inside the box represent the 90% (10%), 75% (25%), and 50% percentiles of the data.

The major drivers that influence the coupling strength of different layers were evaluated through correlation analysis

between environmental properties and the coupling strength. The considered properties were soil properties (the saturated

hydraulic conductivity, bulk density, SOC and soil particle composition), vegetation properties (LAI, NDVI) and precipitation

(mean annual precipitation during 2014-2016) (Fig. 4). Fig. 4 shows that there are significant (p<0.05) but no strong (R<0.45)

linear correlations between precipitation and Lag-R at depth of layer 2 (at depth of 10-20 cm, R=0.33), layer 3 (20-30 cm,

R=0.43), layer 4 (30-50 cm, R=0.33) and the entire profile (0-70 cm, R=0.38). No significant linear correlation was found for

layer 5 (50-70 cm, R=0.29, P=0.09). Also, no linear correlations were found among Lag-R and the other properties (e.g. soil

properties and vegetation properties). Thus, our results indicate that the coupling strength of surface SM and subsurface SM is

mainly controlled by precipitation. At the same time, the positive R values indicate that stations with more precipitation have

stronger coupling strength between surface and subsurface SM (in the top 50 cm of soil and the soil profile of 0-70 cm) in our

study area. Please note that due to lack of measured precipitation data at each soil moisture station, the precipitation in this

study was obtained from a reanalysis dataset (spatial resolution of 0.1°, Chen et al., 2011), which may have introduced

additional uncertainty to our results. Nonetheless, our results are consistent with other studies (e.g. Mahmood et al., 2012;





Wang et al., 2017). For example, a significant positive relationship between precipitation and Lag-R between surface (5 cm) and subsurface (25 cm) SM was reported at Oklahoma, USA (Ford et al., 2014).

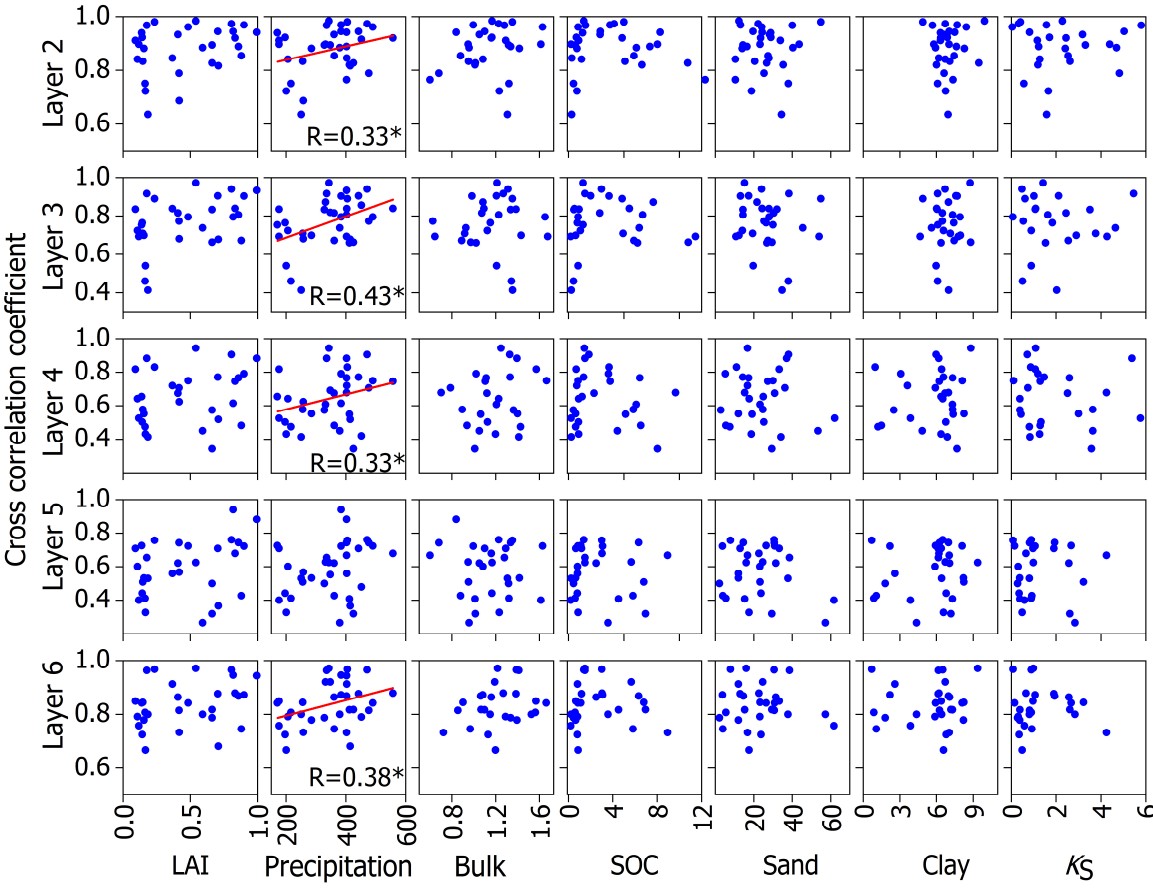


**Fig. 4.** Scatterplot of cross correlation coefficient and the properties of all stations from 2014 to 2016 for different layers. The relationship with significant (p<0.05, *) correlation coefficient is shown in the plot.

## 4.2. Evaluation of different methods

The ExpF method estimates subsurface SM based on SWI, while the ANN and CDF methods are based on volumetric soil

moisture. Following Moriasi et al. (2007), the Nash-Sutcliffe efficiency (NSE), the Ratio of RMSE to the standard deviation of the observations (RSR, an error statistic that normalizes the RMSE), and Pearson correlation coefficient (R) were used to evaluate the performance of different methods with different units. Fig. 5, Table 2 and Fig. S1 summarize the metrics (NSE, RSR, and R) for subsurface SM estimates at different depths obtained using different methods for the growing season of 2014, 2015 and 2016. Fig. 5 shows that there were significant differences for the NSE of different methods for all layers (p<0.05).

The ANN had the highest NSE with a median value of 0.82, 0.56, 0.35, 0.35 and 0.76 for layers 2, 3, 4, 5 and profile SM, respectively. There were no significant differences between the ExpF and the CDF matching method for layer 2, layer 3 and the profile SM. The CDF matching method showed the lowest NSE for layers 4 and 5. Overall, both the ANN and ExpF





methods showed good performance in terms of NSE for layer 2, layer 3 and profile SM (median NSE > 0.5), and the CDF matching method showed a good performance in terms of NSE for layer 2 and profile SM (median NSE > 0.5).

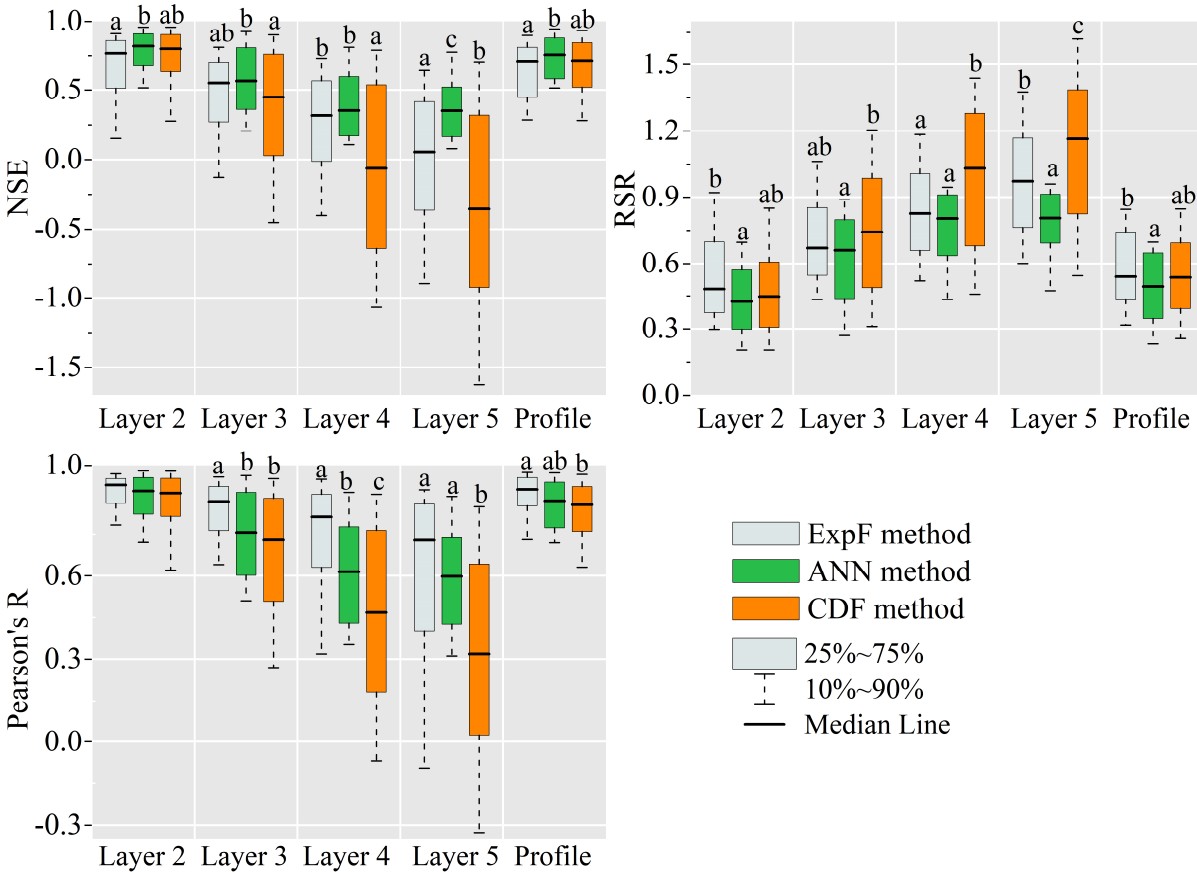

**Fig. 5**. Boxplot of the metrics (NSE, RSR, R) to compare the subsurface SM estimation using surface SM using the three methods (ExpF, ANN, CDF) with the observations for the 35 stations during the growing seasons of 2014 to 2016. Different letters above the box indicate the significant difference (p<0.05) among different methods.

Similar to the results of NSE, Fig. 5 shows that the ANN method resulted in a significantly lower RSR (p<0.01) for all depths. There were no significant differences between the ExpF and CDF matching method for layer 2, layer 3 and profile SM, and the CDF matching method showed a significantly higher RSR for layer 4 and layer 5. The ExpF method had a similar estimation error as the CDF matching method for layer 2, layer 3 and profile SM. Both the ANN and ExpF methods showed satisfactory results (RSR<0.7) for layer 2, layer 3 and profile SM, and the CDF matching method showed satisfactory results for layer 2 and profile SM (Moriasi et al., 2007).

Fig. 5 shows that the ExpF method resulted in the highest R for all layers (with a median value of 0.93, 0.87, 0.81, 0.73 and 0.91 for layers 2, 3, 4, 5 and profile SM, respectively), while the CDF matching method resulted in the lowest R for all layers (with a median value of 0.90, 0.73, 0.47, 0.32 and 0.86 for layers 2, 3, 4, 5 and profile SM, respectively). The good performance for R suggests that the ExpF method had the best ability to describe the temporal variability in SM.



**Table 2.** The statistics (mean±standard deviation) of the performance (RSR, R and NSE) of different methods (ExpF, ANN, and CDF) for estimating subsurface SM using surface SM for each layer of 35 stations during the growing season of 2014, 2015, and 2016

| Layer | RSR | | | R | | | NSE | | |
|---|---|---|---|---|---|---|---|---|---|
| | ExpF | ANN | CDF | ExpF | ANN | CDF | ExpF | ANN | CDF |
| Layer 2 | 0.55±0.25 | 0.44±0.20 | 0.50±0.27 | 0.89±0.10 | 0.87±0.13 | 0.84±0.19 | 0.63±0.36 | 0.76±0.21 | 0.68±0.37 |
| Layer 3 | 0.72±0.27 | 0.62±0.23 | 0.75±0.34 | 0.81±0.19 | 0.74±0.18 | 0.66±0.29 | 0.41±0.50 | 0.56±0.28 | 0.33±0.56 |
| Layer 4 | 0.83±0.27 | 0.75±0.22 | 0.99±0.37 | 0.70±0.31 | 0.61±0.21 | 0.44±0.37 | 0.24±0.47 | 0.40±0.29 | -0.11±0.71 |
| Layer 5 | 0.97±0.29 | 0.77±0.19 | 1.11±0.38 | 0.57±0.39 | 0.58±0.22 | 0.3±0.41 | -0.03±0.61 | 0.37±0.26 | -0.38±0.82 |
| Profile | 0.58±0.22 | 0.49±0.19 | 0.54±0.22 | 0.88±0.11 | 0.85±0.11 | 0.83±0.13 | 0.64±0.32 | 0.73±0.18 | 0.66±0.26 |

As expected, the variation of all the metrics (NSE, RSR and R) with depth corresponded to the vertical variation of the coupling strength as expressed by Lag-R, indicating that the performance of estimation methods was largely determined by the coupling strength between surface and subsurface SM. The results suggest that the ANN method resulted in the lowest estimation error, while the ExpF method was better able to capture SM dynamics. These are similar to Zhang et al. (2017a)'s research, who found that the ExpF method had a significantly higher correlation coefficient along with a higher mean bias compared to the ANN method. For most hydrological researches, the correct temporal variation of SM is more crucial than the exact value, suggesting that more emphasis should be given to R when selecting the most appropriate estimation method (Brocca et al., 2014; González-Zamora et al., 2016). Results also demonstrated that both ANN and ExpF method were able to provide accurate estimates of subsurface SM for layer 2, layer 3 and profile SM, which matches the distribution of the strong coupling strength between surface and subsurface SM.

Overall, the results suggest that the ExpF method is the most useful for estimating subsurface SM from surface SM in our study area. Moreover, the ExpF method is the simplest approach as it has only one free parameter ($T_{opt}$, the optimum characteristic time length), and can thus be easily applied in data-scarce mountainous areas. Therefore, the ExpF method was used to estimate subsurface SM in the remainder of this study.

### 4.3. Evaluation of T*opt* parameter

### 4.3.1. Variation of T*opt* with depth

It was found that the accuracy of the ExpF method varied with the T value, and that higher T values resulted in more stable estimation of SM time series (Wagner et al., 1999; Albergel et al., 2008). Furthermore, they found that each station had an optimum T ($T_{opt}$) as determined based on the best match with observations in terms of NSE. The variation of NSE with T (from 0 to 68 days) for different layers for each station is shown in Fig. 6. It can be seen that the sensitivity of high values of NSE to changes in T decreased with increasing depth, indicating that the range of T values with high NSE was larger deeper in the soil. This was also observed in previous studies (e.g. Wang et al., 2017).





**Fig. 6.** Variation of NSE with T of the exponential filter method at different layers of each stations during the growing season of 2014, 2015 and 2016. Y axis is the NSE value.

As showed in Fig. 7 and summarized in Table 3. Results of a two-way ANOVA (inserted table in Fig. 7 (a)) showed that the difference of $T_{opt}$ is not significant between different years (p>0.05) while significant between layers (p<0.001). Furthermore, it can be seen that $T_{opt}$ increased with depth from layer 2 to 5. The median of $T_{opt}$ ranged from 1.5 for layer 2 to 12.5 days for layer 5. The median $T_{opt}$ for the profile SM was 3.5 days. Significant differences in $T_{opt}$ were obtained for layer 2, 3, and 4, but





the difference between layers 4 and 5 was not significant. The increase of $T_{opt}$ with depth has already been observed in many studies, and is related to the greater temporal stability of SM in the deeper soil layer (Wang et al., 2017; Tian et al., 2019).

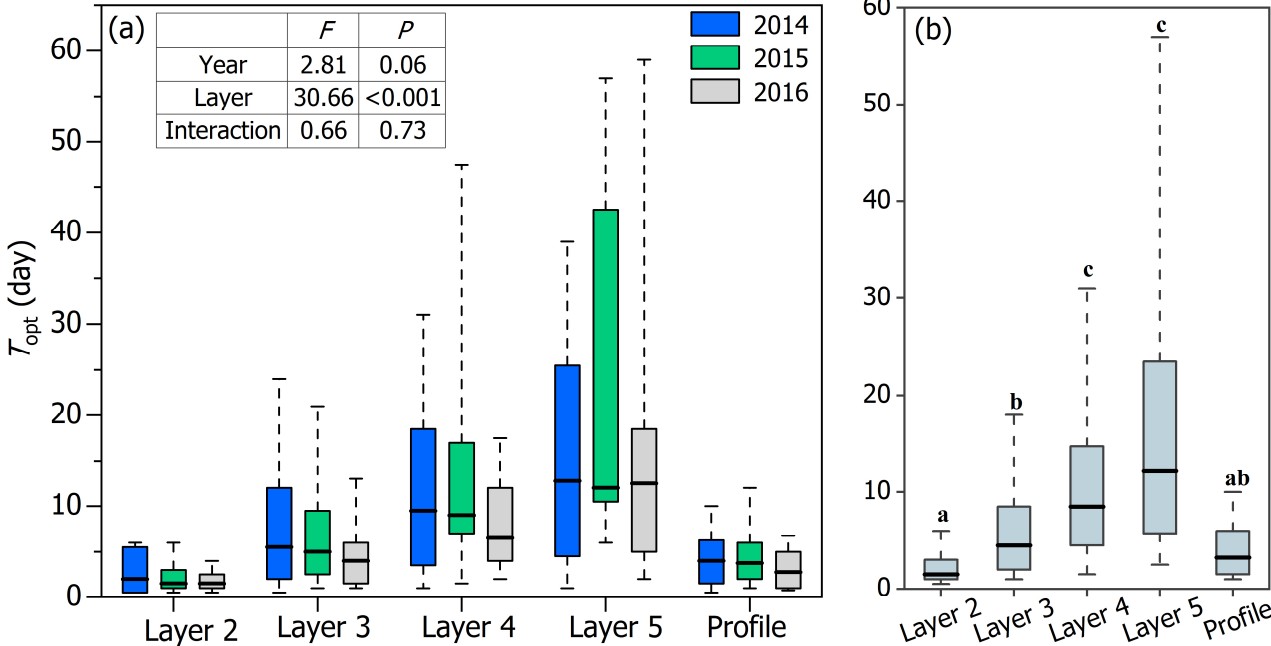

**Fig. 7.** Boxplots to show the variation of $T_{opt}$ with depth at each year (a) and all years (b) for all stations. Two-way ANOVA results (table in (a)) show that the difference of Topt is nonsignificant among years (p>0.05) while significant among layers (p<0.001). Different letters on boxes illustrate significant difference among different layers (p<0.05), while same letter indicates the difference is nonsignificant.

**Table 3.** The statistics of $T_{opt}$ (day) of all stations at different layers of different years

| Year | Statistics | Layer 2 | Layer 3 | Layer 4 | Layer 5 | Profile |
|---|---|---|---|---|---|---|
| 2014 | mean (std) | 2.72 (2.22) | 8.32 (8.39) | 13.18 (12.52) | 16.81 (16.70) | 4.73 (4.16) |
| | median | 2.00 | 5.50 | 9.50 | 12.75 | 4.00 |
| 2015 | mean (std) | 2.56 (2.54) | 7.78 (8.04) | 15.77 (15.87) | 23.15 (19.61) | 5.23 (4.51) |
| | median | 1.50 | 5.00 | 9.00 | 12.00 | 3.75 |
| 2016 | mean (std) | 2.23 (2.13) | 6.13 (9.80) | 9.26 (9.43) | 17.74 (18.93) | 3.32 (2.56) |
| | median | 1.50 | 4.00 | 6.50 | 12.50 | 2.75 |
| Summary | mean (std) | 2.48(2.26) | 7.29(8.85) | 12.37(12.67) | 18.93(18.43) | 4.32(3.77) |
| | median | 1.50 | 4.50 | 8.50 | 12.50 | 3.50 |

Note: std represents the standard deviation, Summary represents the statistics results of the three years.

### 4.3.2 Relationship between $T_{opt}$ and control variables

It is important to understand the factors that control $T_{opt}$. Soil (Rahmati et al., 2018), vegetation (Jia et al., 2017a; 2017b) and climate (McColl et al., 2017) all influence the infiltration process and the dynamics of SM along profile (Rahmati et al., 2018;





Huang and Shao, 2019), thus, the factors are likely to influence $T_{opt}$. Therefore, the relationship between $T_{opt}$ and a series of soil, vegetation, and climate properties was analyzed using linear and nonlinear correlation analysis (including logarithmic, power and exponential relationships). The considered soil properties were the $K_S$, bulk density, SOC and the soil particle composition. In addition, LAI and mean annual precipitation were considered as vegetation and climate properties, respectively. In the following, variables with the highest $R^2$ were selected, and subsequently used in a multiple regression analysis to establish a model to predict $T_{opt}$ for different layers.

The correlation analysis between $T_{opt}$ and different variables is shown in Fig. 8. It can be seen that $T_{opt}$ was negatively correlated with the ln-transformed LAI ($p<0.05$) for all layers, indicating that higher LAI is associated with lower $T_{opt}$ in our study area. Intuitively, a higher LAI suggest a more vigorous vegetation and associated improved infiltration and drainage conditions (e.g. Huang et al., 2019; Sun et al., 2015). Consequently, a stronger temporal variability of subsurface SM as well as a tighter connection between surface and subsurface SM and a lower $T_{opt}$. Fig. 8 also shows that $T_{opt}$ showed a significant negative linear correlation with precipitation ($p<0.01$) for all layers. Stations with higher precipitation experience more frequent infiltration and subsequent drainage (Wang et al., 2015; Ford et al., 2014), which enhance the link between subsurface and surface SM and results in a lower $T_{opt}$. The control of soil properties on $T_{opt}$ was not consistent across all soil depths. For example, bulk density showed significant positive correlations for a power-law relationship ($p<0.05$) with $T_{opt}$ for layer 3 ($R^2=0.24$), layer 4 ($R^2=0.30$), layer 5 ($R^2=0.33$) and profile SM ($R^2=0.30$), but no significant correlation was found for layer 2. Stations with higher bulk density typically exhibit less infiltration, especially for the deeper depth where precipitation has a relatively weak effect, thus leading to a higher $T_{opt}$.

Soil organic carbon (SOC) showed significant ($p<0.05$) and positive correlations for an exponential relationship with $T_{opt}$ for layer 2 ($R^2=0.21$), layer 3 ($R^2=0.42$), layer 4 ($R^2=0.24$), and profile SM ($R^2=0.23$). No significant correlation was found for layer 5. Soils with higher SOC facilitate water infiltration in the study area (Yang et al., 2014), which results in a lower $T_{opt}$. A significant linear correlation between $K_s$ and $T_{opt}$ was only found for profile SM ($R^2=0.23$). At the same time, the ln-transformed clay fraction showed a significant relationship with $T_{opt}$ for layer 2 ($R^2=0.20$) and layer 4 ($R^2=0.16$).

Previous studies have shown conflicting results with respect to the control of soil properties on $T_{opt}$. For example, a strong correlation among soil particle compositions and $T_{opt}$ for profile SM (0-100 cm) was found through both the observations (Ceballos et al., 2005) and simulations (Lange et al., 2008). While nonsignificant relationships between soil texture and $T_{opt}$ was also reported based on the previous researches (e.g. Albergel et al., 2008; Wang et al., 2017).

In a next step, stepwise multiple regression was used to establish a model to predict $T_{opt}$ from the variables identified in the correlation analysis described above. Standardized regression coefficients were used to remove the influence of the different units of the independent variables and to allow an analysis of the relative contribution of the control variables (Nathans et al., 2012). The regression equations were significant ($p<0.01$) for all layers and the adjusted $R^2$ was higher than 0.46 for all layers (Table 4). The variables that were selected in the stepwise regression were quite different among layers. The standardized regression coefficients indicated that the most important predictor was precipitation for the upper layers (layers 2 and 3), whereas LAI was more important for the deeper layers (layer 4, layer 5 and profile SM).

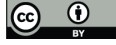

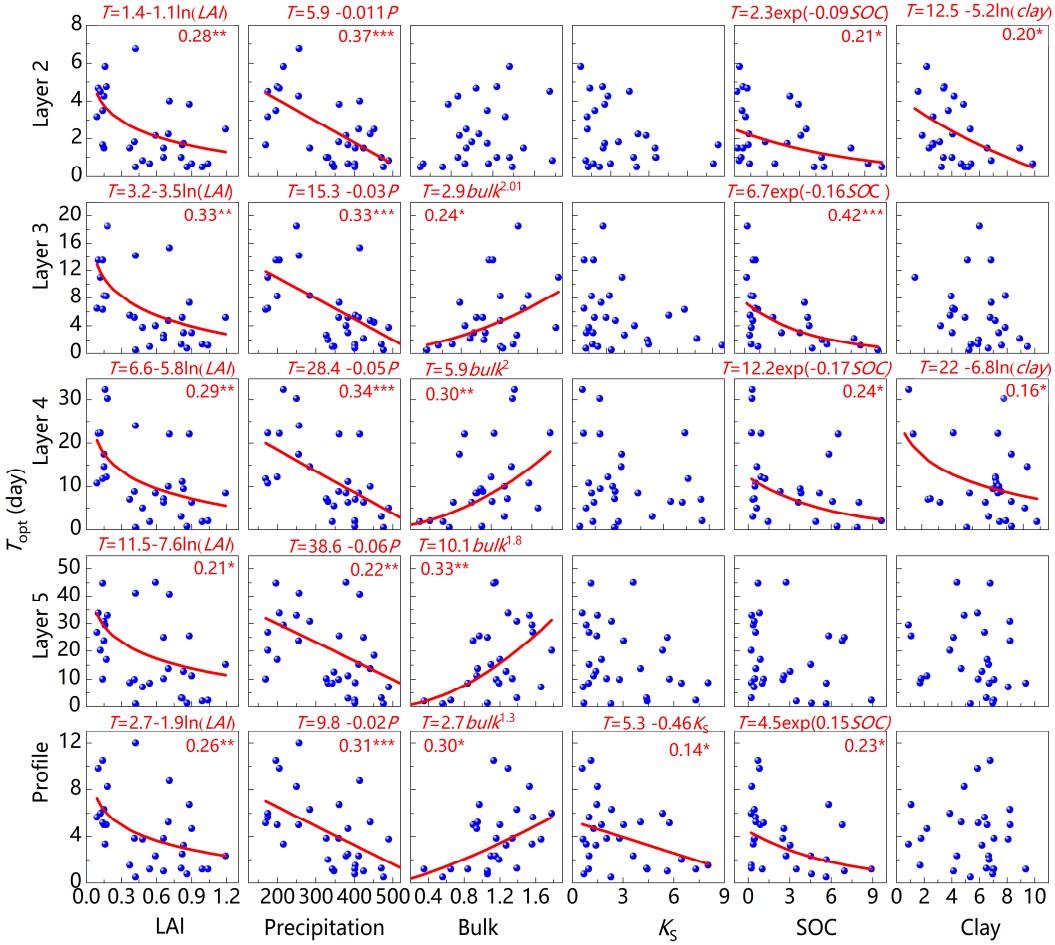

**Fig. 8.** Scatter plot and the fitted curve between the $T_{opt}$ and variables of specific layer. The y axis is the averaged $T_{opt}$ for specific layers of each station from 2014 to 2016. The red curve and equation of each graph represent the best relationship between $T_{opt}$ and variables at the statistically significant level (p<0.05), and the number in each graph represents the $R^2$. *, **, and *** represents significant level at <0.05, <0.01, and <0.001, respectively. LAI, Bulk, Precipitation, $K_S$, SOC and Clay represent the leaf area index, soil bulk density (g cm$^{-3}$), the annual precipitation (mm), soil saturated hydraulic conductivity (cm/hour), soil organic carbon (g/100g), and the clay percent (%), respectively.

**Table 4.** The results of multiple regression equations for the relationship between $T_{opt}$ (day) and soil, vegetation and climate properties for different layers.

| Layer | Equation | R | Adjusted $R^2$ |
|---|---|---|---|
| Layer 2 | $T_{opt} = 13.71 - 0.11 \times Prec - 3.90 \times \ln(clay)$ | 0.83*** | 0.66 |
| Layer 3 | $T_{opt} = 14.91 - 0.23 \times Prec - 1.35 \times e^{(SOC)}$ | 0.79*** | 0.59 |
| Layer 4 | $T_{opt} = 18.55 - 6.98 \times \ln(LAI) - 8.28 \times \ln(clay)$ | 0.79*** | 0.59 |
| Layer 5 | $T_{opt} = 20.17 - 11.38 \times \ln(LAI) - 5.17 \times \ln(sand)$ | 0.78*** | 0.56 |
| Profile | $T_{opt} = 4.64 - 2.44 \times \ln(LAI) - 1.03 \times \ln\ln(sand)$ | 0.71** | 0.46 |

Note: ** and *** indicate that the correlation is significant at p < 0.01 and p < 0.001, respectively.





### 4.3.3 Evaluation of alternative methods for $T_{opt}$ estimation

Previous studies have used different methods to estimate $T_{opt}$ for a study area of interest. For example, it was found that using a single representative value for $T_{opt}$ (e.g. average or median) for all stations did not significantly reduce the accuracy of the SM estimates (Albergel et al., 2008; Ford et al., 2014). Alternatively, Wagner et al. (1999) recommended a common value of

$T_{opt} = 20$ (days) to estimate the root zone SM, and this value has been widely adopted (e.g. Lange et al., 2008; Muhammad et al., 2017). Additionally, Qiu et al. (2014) proposed to estimate $T_{opt}$ using the station-specific long-term mean NDVI using $T_{opt} = -75.263 \times \text{NDVI} + 68.171$ (R=0.5, p<0.01). This approach has also been applied in other studies (Tobin et al., 2017).

Here, we evaluated five different methods to estimate $T_{opt}$ in our study region by initializing the ExpF method for estimating profile soil moisture (0-70 cm, SWI) from surface soil moisture (5 cm, SWI). In the first method, $T_{opt}$ was estimated from the

NDVI-based regression of Qiu et al. (2014) to provide $T_{Qiu}$. In the second method, $T_{opt}$ was set to 20 days as recommended by Wagner et al. (1999) to provide $T_{Wagner}$. In the third method, an area-generalized $T_{opt}$ was obtained from the median value for the profile SM in our study region (3.5 days) to provide $T_{general}$. In the fourth method, $T_{opt}$ was estimated from the regression equation for profile SM using the leave-one-out method. For this, we estimated the regression equation based on the variables (station-specific $T_{opt}$, ln-transformed LAI and ln-transformed sand according to Table 4) of 34 stations and predicted the $T_{opt}$

for the remaining station one at a time, and repeated it 35 times to obtain the estimated $T_{opt}$ at 35 stations, which are named $T_{regression}$ in the following. In the fifth and final method, the original station-specific $T_{opt}$ parameter for profile SM was used and named $T_{specific}$.

The accuracy of the SM estimates obtained using the different methods to estimate $T_{opt}$ was evaluated using the NSE, R and RMSE (Fig. 9). As expected, the original $T_{specific}$ values performed best (mean RMSE of 0.13, R of 0.88, and NSE of 0.64).

However, $T_{regression}$ and $T_{general}$ also provided low RMSE and high R and NSE values that were not significantly different from those obtained with $T_{specific}$. The $T_{Wagner}$ (mean RMSE of 0.19, R of 0.69, and NSE of 0.32) and the $T_{Qiu}$ approach (mean RMSE of 0.22, R of 0.59, and NSE of 0.17) performed less good, and the metrics were significantly (p<0.001) lower than those of the $T_{general}$, $T_{regression}$ and $T_{specific}$ methods. Our results suggest that a site-specific $T_{opt}$ significantly improves the performance of the ExpF method compared to the use of the universal $T_{opt}$ recommended by Wagner et al. (1999) or the regression of Qiu et

al. (2014). Similarly, Lange et al. (2008) also found a significant improvement when using a station-specific $T_{opt}$ instead of $T_{opt}$ = 20 days. It should be mentioned that the estimation depth in the method of Wagner et al. (1999) was 0-100 cm, while that of our study was 0 - 70 cm. This may partly explain the poor performance of the $T_{Wagner}$ approach in this study. The use of an area-generalized $T_{opt}$ (3.5 days) or the multiple regression approach are suitable alternatives for $T_{opt}$ estimation in our study area, and provide similar estimation performance. Other studies have also found a good performance when using an area-generalized

$T_{opt}$ (e.g. Albergel et al., 2008; Brocca et al., 2010; Ford et al., 2014).





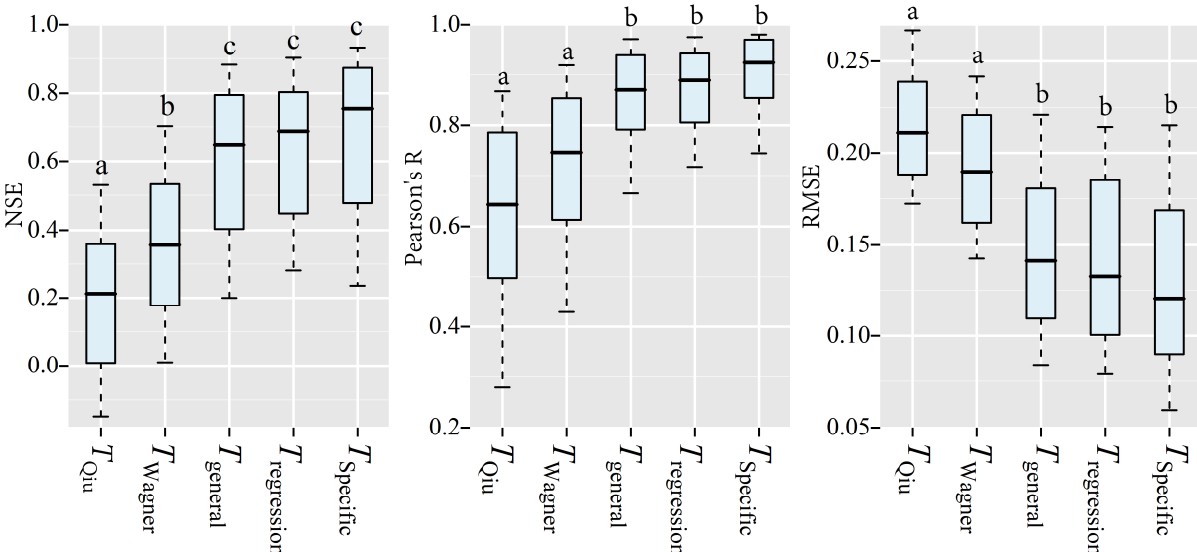

**Fig. 9.** The boxplot of NSE, Pearson's R, and RMSE for the $T_{opt}$ generated from different schemes. The different letters above box indicate the significant difference for different schemes.

## 4.4 Estimating profile soil moisture using SMAP

The results of the cross-correlation analysis showed that surface SM is strongly coupled with profile SM, and it was also found that the ExpF method is most suitable to estimate profile SM from surface SM. In this final section, we evaluate the utility of the ExpF method in combination with SMAP surface products for estimating profile SM in mountainous areas.

### 4.4.1 Assessment of SMAP surface SM product

The observed surface SM of each station was compared with the SMAP_L3 soil moisture product that overlapped with the
corresponding station for the growing seasons of 2015 and 2016 for all stations to evaluate the accuracy of the SMAP measurements (Pablos et al., 2018). The root mean square error (RMSE), mean bias error (MBE), unbiased RMSE (ubRMSE) and R were adopted as metrics to evaluate accuracy. A scatterplot of SMAP-based SM against in-situ SM for all stations is shown in Fig.10. Time series of the two datasets for each station are shown in the supplementary Fig. S2. Figs 10 and S2 show that the performance was low at two stations (D13 with R of 0.18, D15 with R of 0.08) with scrubland and relatively high soil
moisture. The poor performance at scrubland sites persistent with Zhang et al. (2017b)'s researches in this study region. Results showed that the MBE varied from -0.23 to 0.07 cm³/cm³ with median of -0.021 cm³/cm³. This indicates that SMAP underestimated surface SM over the study region, which is consistent with previous studies in the area (Chen et al., 2017; Zhang et al., 2017b). And RMSE varied within 0.026-0.250 cm³/cm³ between sites with a median value of 0.052 cm³/cm³. After removing the bias, the SMAP product had a median ubRMSE of 0.036 cm³/cm³ (range from 0.024 to 0.083 cm³/cm³).
Therefore, the SMAP product achieved the accuracy requirement of 0.04 cm³/cm³ (Chan et al., 2016) in this study area. The R value ranged from 0.075 to 0.81 with a median value of 0.59. The relationship between SMAP-derived and in-situ observed





surface SM was significant (p<0.05) at all but one station. This suggests that the SMAP surface product can represent the temporal dynamics of the observed surface SM time series.

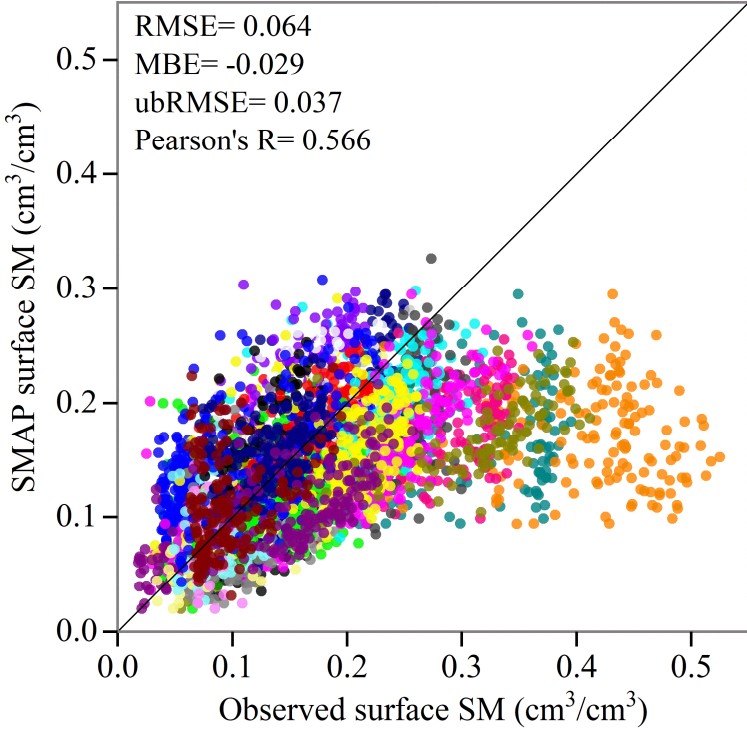

**Fig. 10.** Scatterplot of the SMAP_L3 surface SM (cm³/cm³) with in-situ observations at the surface (5 cm) of the 35 soil moisture stations. Each color indicated one station. Averaged metrics (RMSE, MBE, R, ubRMSE) of 35 stations during the growing seasons of 2015 and 2016 were showed in the plot.

**4.4.2 SMAP-based estimation of subsurface soil moisture**

The SMAP surface soil moisture was combined with the ExpF method to estimate the subsurface SWI (Layer 2: 10-20 cm, Layer 3: 20-30 cm, Layer 4: 30-50 cm, Layer 5: 50-70 cm, Profile: 0-70 cm) during the growing season of 2015-2016. The site-specific $T_{opt}$ for SMAP product was determined based on the best match between observations in terms of NSE. Finally, daily estimates of SMAP-derived subsurface SWI were compared to the in-situ subsurface SWI at each station (time series of the comparison at different layers in supplementary Fig. S3-S7, scatterplot between measured and predicted SWI in Fig. 11), and the statistics for the metrics of the comparisons are summarized in Table 5.

As expected, the estimation accuracy of subsurface SM decreased with depth, which coincided with the variation of coupling strength. The ANOVA results showed that the subsurface SM estimation accuracy at layer 2 (median value of RMSE=0.185, R=0.729) and profile (RMSE=0.192, R=0.718) were higher than that of layer 4 (RMSE=0.217, R=0.524) and layer 5 (RMSE=0.236, R=0.546) significantly (p<0.05). Meanwhile, the negative MBE showed that the estimations underestimated the subsurface SM. Moreover, the relationship between SMAP-derived and in-situ observed subsurface SM at both layer 2 and



profile was significant (p<0.01) at all stations but one station (D15). Thus, the SMAP surface product and ExpF method can

be used to estimate the subsurface SM in the study area, especially for the layer 2 (10-20 cm) and profile (0-70 cm) SM.

**Fig. 11.** Scatterplot of the comparisons of SMAP_L3 estimated-observed subsurface SWI for all stations during the growing seasons of 2015-2016. The representation is a smoothed color density of a scatter plot to make a quantity of points visual. The dash and solid line are
the best-fitted curve and "y=x" line, respectively.

As suggested by Ford et al. (2014), we partition the error in the SMAP-based estimation of profile SWI ("SMAP-observed profile SWI", Fig. S8(c)) in errors associated with the ExpF method and errors due to SMAP observation differences to get insight into the major source of error in SMAP-based estimates of profile SWI. For this, profile SWI estimated using the ExpF method from observed surface SWI was compared with in-situ observed profile SWI ("estimated-observed profile SWI") to
assess errors of the ExpF method (Fig. S8(a)). In addition, SMAP-based and in-situ observed surface SWI ("SMAP-observed surface SWI") were compared to assess inherent errors of the SMAP product (Fig. S8 (b)). RMSE, R and MAE were used as the metrics to assess accuracy. Results of the analysis are shown in Fig. S9 and summarized in Table 6.

Fig.S9 and Table 6 show that the SMAP-observed SWI comparisons had lower performance metrics for surface SWI (median R of 0.581, median MAE of 0.177) than for profile SWI (median R of 0.634, median MAE of 0.159). This may be



because the profile SWI was estimated based on the SMAP surface SWI and $T_{opt}$, which was determined by optimization using the maximum NSE, thus, improved the performance of estimation. In addition, the performance metrics for SMAP–observed SWI comparisons for both surface and profile SWI were significantly (p<0.001) lower than those of estimated–observed profile SWI (with a median R of 0.881, MAE of 0.11). Thus, the major error in SMAP-based profile SWI estimates stems from the SMAP satellite product and is not derived from ExpF method, which is also supported by other researches (e.g. Ford et al.,

2014; Pablos et al., 2018). Notably, the SMAP_L3 product has a spatial posting of 9 km×9 km, while the in-situ measurements are point-based and soil moisture has a strong spatial variability in mountainous areas (Tian et al., 2019). Thus, the disparity of spatial scales between points and satellite footprints will introduce additional errors in the validation of the satellite products (Jin et al. 2017).

**Table 5.** Statistics of the metrics (RMSE, R, MAE) of the comparisons of SMAP estimated and observed SWI at different layers for the 35 stations during the growing season of 2015-2016.

| Layer | RMSE | | R | | MAE | |
|---|---|---|---|---|---|---|
| | Mean±Std | Median | Mean±Std | Median | Mean±Std | Median |
| Layer 2 | 0.199±0.056 | 0.185a | 0.644±0.241 | 0.729a | 0.167±0.053 | 0.156a |
| Layer 3 | 0.214±0.062 | 0.207ab | 0.547±0.307 | 0.643ab | 0.179±0.057 | 0.174ab |
| Layer 4 | 0.232±0.075 | 0.217b | 0.429±0.378 | 0.524b | 0.195±0.066 | 0.189b |
| Layer 5 | 0.253±0.088 | 0.236b | 0.412±0.391 | 0.546b | 0.213±0.077 | 0.206b |
| Profile | 0.187±0.054 | 0.192a | 0.634±0.269 | 0.718a | 0.159±0.048 | 0.164a |

Note: the different letters after the number indicate that the difference are significant at p<0.05 (Kruskal-Wallis ANOVA)

**Table 6.** Statistics of the metrics (RMSE, R, MAE) of the comparisons of estimated-observed profile SWI datasets, SMAP_L3-observed surface SWI datasets, SMAP_L3-observed profile SWI datasets, and SMAP_L4-observed profile SWI datasets for the 35 stations during the growing season of 2015-2016.

| Comparisons | RMSE | | R | | MAE | |
|---|---|---|---|---|---|---|
| | Mean±std | Median | Mean±std | Median | Mean±std | Median |
| Estimated-observed PSWI | 0.129±0.057 | 0.125 | 0.881±0.109 | 0.899 | 0.110±0.053 | 0.102 |
| SMAP_L3-observed SSWI | 0.221±0.062 | 0.209 | 0.581±0.192 | 0.641 | 0.177±0.058 | 0.166 |
| SMAP_L3-observed PSWI | 0.187±0.054 | 0.192 | 0.634±0.269 | 0.718 | 0.159±0.048 | 0.164 |
| SMAP_L4-observed PSWI | 0.243±0.088 | 0.241 | 0.470±0.313 | 0.553 | 0.204±0.079 | 0.201 |

Note: e.g. Estimated-observed PSWI means the comparison of the estimated profile SWI and observed profile SWI.

Our study showed that profile soil moisture can be estimated from the SMAP_L3 surface product. In addition, the SMAP mission also provides the L4 SM products with both surface and root zone at depth of 0-5 cm and 0-100 cm, respectively (Reichle et al., 2017), which is a widely used root zone SM product (Pablos et al., 2018). For comparing our results with the SMAP_L4 root zone product, the SMAP_L4 SM products with both surface ($sm_{0-5}$) and root zone ($sm_{0-100}$) were used to

calculate the SM at 0-70 cm profile ($sm_{0-70}$) using:

$$sm_{0-100} = (5 * sm_{0-5} + 95 * sm_{5-100})/100 \tag{8}$$





$$sm_{0-70} = (5 * sm_{0-5} + 65 * sm_{5-100})/70 \tag{9}$$

Subsequently, the SMAP_L4 and SMAP_L3 estimated profile SWI were compared to the in situ observed profile SWI (see Fig. S10 and Table 6). From these results, we can see that the performance of soil profile moisture estimation using the SMAP_L3 surface product and the ExpF method (mean RMSE, R and MAE of 0.187, 0.634 and 0.159, respectively) was significantly (p<0.01) higher than that of the SMAP_L4 product (mean RMSE, R and MAE of 0.243, 0.470 and 0.204, respectively). The low performance of SMAP-L4 profile product may be also caused by the meteorological driving forces and the soil parameters in the NASA catchment model are not very appropriate for the cold mountain area (Reichle et al., 2017; Zhao et al., 2018; Dai et al., 2019). Thus, result suggests that combining Exponential filter method with the SMAP_L3 improves the estimation of profile SM for the data-scarce cold arid mountainous areas.

Moreover, we tested whether the ExpF method and the derived parameters are robust around the Qilian Mountain region. To this end, we used SM data of the Maqu observation network located at the Qinghai-Tibet plateau (Dente et al., 2012; Su et al., 2013; Su et al., 2011; van der Velde et al., 2012; Zeng et al., 2016) (Fig. S11). We initialized the ExpF method for estimating profile SM (0-70 cm) from surface SM (5 cm) using both the site-specific $T_{opt}$ of the Maqu SM stations ($T_{specific}$) and the median value of $T_{opt}$ in the Heihe SM network ($T_{heihe}$=3.5 days). The comparisons of the estimations ($T_{specific}$ and $T_{heihe}$) with observations of SWI time series are shown in Fig. S12. Furthermore, the statistical metrics (NSE, Pearson's R, RMSE and MAE) of the estimation using different T are presented in Fig. S12, Fig. S13 and Table S1.

Our results demonstrate that the ExpF method also showed a good performance in estimating profile SM from surface SM for Maqu SM network. Using the site-specific $T_{opt}$, the average RMSE, R and NSE were 0.09, 0.95 and 0.86, respectively. In addition, the performance of $T_{heihe}$ (average RMSE, R and NSE of the estimation were 0.11, 0.94 and 0.81, respectively) was not significantly ($P_{ANOVA}$=0.55) lower than that of $T_{specific}$. These results demonstrate that both the ExpF method as well as the parameter ($T$) that we obtained from the Heihe network ($T_{heihe}$) provide robust estimates in other areas of the Qilian mountainous region, and not only within the Heihe River Basin.

Finally, for the purpose of getting the spatial distribution of profile soil moisture for a large-scale mountainous area, we interpolated the site-specific profile $T_{opt}$ to get the spatial distribution of profile $T_{opt}$ (Fig. S14). Then, the spatially distributed $T_{opt}$ was combined with the SMAP_L3 product to get the spatial distribution of profile SM in the study area during the growing season of 2015, 2016 and 2017 (Fig. 12). The spatial distribution of the profile SM shows a high value at the southeast, while low value at the northwest of the study area, which was coincided with the spatial distribution of the precipitation and surface SM. The temporal variation of the profile SM shows an increase from May to September and then decreases during the growing season.

Previous studies have shown the difficulty of applying the ExpF method to satellite products in mountainous area, where complex topography (Paulik et al., 2014), snow and soil freezing (Ford et al., 2014; Pablos et al., 2018) cause large errors and poor performance of the filtering method (Albergel et al., 2008). Ford et al. (2014) found an improvement of performance after removing the effects of snow from the data in the SCAN network, USA. Based on in situ SM observations, this study showed

that the ExpF method is useful in estimating profile SM from SMAP surface products in the growing season in high and cold

mountainous areas.



**Fig. 12.** The spatial distribution of the monthly averaged profile SWI product estimated from SMAP_L3 surface product during the growing season from 2015 to 2017.

**5. Conclusions**

We evaluated the coupling strength between surface SM (0-10 cm) and subsurface SM (10-20, 20-30, 30-50, 50-70, 0-70 cm)

using a 3-year dataset from a large-scale SM observation network installed in the upstream mountainous areas of the Heihe

River Basin. In addition, this study tested the performance of three methods (ExpF, ANN, CDF) for estimating subsurface SM

from surface SM, and finally evaluated the utility of the ExpF method to estimate profile SM from SMAP surface products in

the study area. main findings are:



1) A strong coupling exists between surface and subsurface SM within the top 70 cm of the soil in our study area, which enables a reliable estimation of subsurface SM from surface SM.

2) The accuracy of the three methods for profile SM estimation decreased with depth, and this matched the variation of coupling strength.

3) Both the ANN and ExpF methods showed good performance for the estimation of SM at depth within the top 30 cm of soil and profile SM from surface SM measurements at 5 cm. The ANN method exhibited the lowest estimation error, while the ExpF method was able to better capture the temporal variation of subsurface SM.

4) The area-generalized $T_{opt}$ value of the ExpF method can be used in the study area to estimate the subsurface SM without significantly reducing the performance compared to the station-specific $T_{opt}$.

5) The ExpF method was able to estimate profile SM from SMAP surface products with reasonable accuracy (median R of 0.718). In addition, we found that the combination of the ExpF method and the SMAP_L3 surface product can significantly improve the estimation of profile soil moisture in cold arid mountainous areas (e.g. compared to the SMAP_L4 root zone product).

We anticipate that our findings can improve the estimation of subsurface SM at large-scale mountainous areas, which in turn
will support ecohydrological research and water resources management in inland river (terminal lake) basins.

*Data availability*. All the data used in this research are available upon request.

*Author contributions*. JT, BZ and CH prepared the research project. JZ, BZ, CH, HB and JH conceptualized the methodology. JT, BZ and CM collected the data. JT developed the code and performed the analysis. JT prepared the manuscript with contributions from all co-authors.

*Competing interests*. The authors declare that they have no conflict of interest

**Acknowledgements:**

The project is partially funded by the National Natural Science Foundation of China (grants 41530752, 51609111 and 91125010) and Fundamental Research Funds for the Central Universities (lzujbky-2016-256). We are grateful to the members of the Center for Dryland Water Resources Research and Watershed Science, Lanzhou University for their efforts to collect
the soil moisture data and maintain the stations in this high, cold, and inaccessible mountainous area. Without their hard work, the soil moisture data presented in this paper would not have been available. We also thank the National Tibetan Plateau Data Centre (https://data.tpdc.ac.cn/en/) for providing supporting data. The first author also wishes to express his appreciation for the assistance and friendship that he experienced during his stay at the Forschungszentrum Jülich from September 2017 to March 2019.



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
