# Peer review of "Estimation of subsurface soil moisture from surface soil moisture in cold mountainous areas"

_Hydrology and Earth System Sciences, 2019_

## Referee Comment (RC1) · Anonymous Referee #1 · 17 Dec 2019

The paper explores methods to estimate subsurface soil moisture from surface soil moisture based on an in-situ observations in cold mountainous areas since 2013. This variable is important for different scientific and applied topics. Authors explored the availability of three methods and applied the exponential filter method to the SMAP product. Research showed the improvement of profile soil moisture estimations in the mountainous. Many useful data, figures, and results were shown in the manuscript. I recommend a minor revision.

General comments: The paper is well written and the results are well presented. Bibliography very exhaustive. The analyzed dataset is interesting and the results can be useful to improve the estimation of subsurface soil moisture and could be potentially useful for hydrological modelling. The results show that the combination of exponential filter method and satellite surface product can improve the estimation of profile soil moisture, and the availability of the area-generalized Topt in the cold mountainous areas. Related researches in high mountain ranges are limited around the world. Therefore, the presented results add new knowledge on those relevant hydrologic topics. 1. Line 111, The half-hourly measurements were averaged to obtain daily SM values that will be used for the estimation of subsurface SM, which cover up the response of soil moisture to precipitation in a day if it's a rainstorm in where are a big soil porosity. 2. Figure 12, please explain this figure in detail about the temporal variation of soil moisture. It's obvious that SM increased in August. You can link the impact of climate change to moisture source and so on.

Specific comments: Line number are related to the authors' line numbers. 3. Line 14, 'statistical' replace with 'multiple'. 4. Line 15, 'an' replace with 'its'. 5. Line 15, '15' and '25' replace with '10-20' and '20-30', respectively. 6. Line 25, 'with' replace with 'by'. 7. Line 26, please rewrite this sentence. I would prefer to 'the ExpF method was applied to estimate profile soil moisture using the satellite soil moisture product'. 8. Line 27, the first 'to' replace with 'with'. 9. Line 33, please insert 'as' before 'an'. 10. Line 36, 'included' replace with 'include'. 11. Line 41, 'provide' replace with 'provides'. 12. Line 48, 'from 0 to 60 cm depth' replace with '(from 0 to 60 cm depth) '. 13. Line 56, please delete 'that'. 14. Line 59, 'are' replace with 'is'. 15. Line 60, 'have' replace with 'has'. 16. Line 61, 'on' replace with 'about'. 17. Line 73-74, please re-write this sentence 'In the absence of large scale networks of in situ SM observations in mountainous areas'. 18. Line 84, delete 'which is'. 19. Line 106, please put the reference 'Zhang et al., 2017b' at the end of the sentence. 20. Line 107, ' (González-Zamora et al., 2016) ' replace with 'González-Zamora et al. (2016) '. 21. Line 109, 'data set' replace with 'dataset'. 22. Line 118, delete the 'and' after 'bulk density'. 23. Line 123, I think you use the SMAP products of 2015-2017 in your research, not only 2015-2016. 24. Line 118, delete 'and' after 'bulk density'. 25. Line 155, 'tn-1' replace with 'tn-1'. 26. Line 166, delete ',' before 'The ANN'. 27. Line 167, delete 'of' after 'training'. 28. Line 177, I think the equation (6) is incorrect, please correct it. 29. Line 190, please unify the 'lag

time' and 'Lag time', I think it's better to use the term 'Lag time'. 30. Line 193, 'from 0-70 cm' replace with '(from 0-70 cm)'. 31. Line 204, add ','respectively at the end. 32. Line 211, I think you mean that 'no significant linear correlations' rather than 'no linear correlations'. 33. Line 216, delete 'have' after 'may'. 34. Line 250, 'season' replace with 'seasons'. 35. Line 270, insert 'ranging' before 'from'. 36. Line 280, please insert 'layer' before both the '3' and '4'. 37. Line 285, 'Topt' replace with 'Topt'. 38. Line 300, 'suggest' replace with 'suggests'. 39. Line 310, I think it is negative correlations from Fig. 8. 40. Line 356, 'Topt' replace with ' Topt '. 41. Line 380, insert 'were shown' before both 'in supplementary' and 'in Fig.11'. 42. Line 380, 'researches' replace with 'research'. 43. Line 420, 'Topt' replace with 'Topt'. 44. Line 421, insert 'profile SWI' before 'estimation'. 45. Line 430, 'season' replace with 'seasons'. 46. Line 444, 'soil profile moisture' replace with 'profile soil moisture'. 47. Line 447, 'SMAP-L4' replace with 'SMAP_L4'. 48. Line 485, insert 'The' before 'main findings'.

―――――――――――――――――

---

## Short Comment (SC1) · 22 Dec 2019

1. The aim of this study is ambiguous. Is it comparison of different methods, improvement of methods, or evaluation of satellite products? 2. Does the ExpF method with optimum Topt perform better than the ANN? If not, why does the author apply the ExpF method to expand SMAP? 3. In the introduction, the author mentions there are four groups of methods, what are their advantages and disadvantages? Why did the author choose the three methods in this study? 4. In section 4.1, there are lag time between soil moisture data at different layers and at surface. How did the author consider the impacts of the lag time in applications of these mehtods?' 5. The performance of the ANN method is significantly related to the training data. In this study, 70 % data was used as training data acording to Zhang's study. However, Zhang's study focused on

the US., is 70 % suitable for the high mountianous area? Moreover, even with a ratio of 70%, there are lots of data combinations, what's the pricinple to choose these data?Does the author compare the performance of the ANN method with different data combinations? 6. In section 4.3.2, Topt is estimated by precipitation and clay ratio. However, the main advantage of the RBF method is its requirement of few data in introduction. Thus, the improvement in this study is meaningless. What's the insight of this improvement in other regions? 7. The author evaluates both SMAP_L3 and SMAP_L4 products against in situ observations. The SMAP_L4 is the assimilation results of satellite data and model simulation. What's the impacts of their original biases of SMAP_L3 and SMAP_L4 respectively? What's the impacts of scale-mismatch between footprint scale of satellite products and point scale of in situ observations?

---

## Referee Comment (RC2) · Anonymous Referee #2 · 7 Jan 2020

General comments: This manuscript describes a wide-variety of approaches to estimate subsurface and profile soil moisture from surface soil moisture data in the Qilian Mountains of China. Most of the conclusions are well-supported, but the manuscript suffers from statistical inconsistencies. A consistent set of statistical measures should be maintained throughout the manuscript (NSE, RSR, and R). The only exception could be Fig. 10 where the aim is to compare the SMAP ubRMSE to the mission's accuracy requirement. Also, 30% of the data should be withheld for validation for all three methods, not only the ANN method. The performance of the SMAP-based ExpF method for estimating profile soil moisture (i.e., SWI) is overstated, and including the NSE and RSR statistics will likely provide a much more objective view. The manuscript also suffers from poor organization in some places and is not well-written.

[Figure]

Specific comments: 1. line 126. Clarify the meaning of "to consider the assumption of uniform vertical profiles of soil temperature and soil dielectric properties".

2. Table 1. Results are being reported with too many significant figures. I doubt that the laboratory is able to measure sand and silt content with adequate precision to justify four significant figures. Reduce to a more appropriate level, perhaps three significant figures.

3. Section 3.4. Provide more explanation. Was a separate ANN model developed for every depth combination and every site?

4. Equation 6 is the wrong equation.

5. line 196. not "persistent" but "consistent"

6. line 205-208. This should be moved to the methods section.

7. line 257. You have not provided any convincing evidence that "For most hydrological researches, the correct temporal variation of SM is more crucial than the exact value, suggesting that more emphasis should be given to R when selecting the most appropriate estimation method." You have presented three statistical measures to evaluate these methods (RSR, R, and NSE). For two out of the three statistical measures (RSR and NSE) the ANN method had the best performance. Therefore, you should include a clear statement that the results from the ANN method were statistically superior to those from the other two methods. You are still free to prefer the ExpF approach if it is simpler to apply than the ANN method. Just don't try to justify that choice on a statistical basis.

8. Figure 7a is unnecessary and should be deleted. Your results show that "Year" does not have a significant effect, so the data should be presented including all years as done in Fig. 7b.

9. line 290-297. This should be moved to the methods section.

10. line 298-305. This section is not convincing. How strong is the correlation between ln-transformed LAI and precipitation? Perhaps the apparent relationship between Topt and LAI is a spurious result of the correlation between LAI and precipitation.

11. line 319-322. Move to methods.

12. line 319-322. What steps were taken to prevent problems due to collinearity of the predictor variables?

13. Table 4. Is "ln ln (sand)" correct in the last row?

14. Fig. 9. Present RSR instead of RMSE to be consistent with the rest of the manuscript.

15. line 380. Not "persistent" but "consistent".

16. Section 4.4.1. You should note an important limitation of this analysis. There is a huge scale mismatch between the 9 km SMAP data and the in situ sensors which measure at a single point. This will likely degrade the agreement between the two data sets.

17. line 394-399. Move this to methods.

18. line 394-399. Why did you even bother all the effort to determine Topt from the in situ stations in the prior sections? Now you are not using those Topt values but instead finding new ones based on comparison of the SMAP data with the in situ data. This does not make sense in the flow of the manuscript.

19. line 400-406. Include NSE and RSR measures here. They are crucial for quantifying the mismatch between the SMAP SWI and the observed SWI values as shown in Fig. 11.

20. line 425-428. This point should also have been made in Section 4.4.1.

21. Tables 5 and 6. Replace RMSE with RSR. Add NSE.

22. line 436. Your results (Fig. 11) show that the performance of the SMAP profile SWI estimates is relatively poor. This is being partly obscured by the omission of the NSE and RSR statistics.

23. line 451-452. Here on the 22nd page of the manuscript a completely new data set is introduced. This is inappropriate. If this section is important to the manuscript, then take the time to justify it in the introduction and describe it in the methods.

24. line 465. Interpolated how? What evidence do you have that the interpolation is statistically valid? What is the associated uncertainty? This again should be justified in the introduction and described in the methods.

25. line 465. Also, why bother to spatially interpolate Topt? You have just argued that Topt defined in one region (Heihe) is valid in another region (Maqu).

26. line 495. The data in Fig. 11 show that the accuracy is relatively poor. Relying on the R value alone is clearly misleading in this case where there is a substantial bias. Including the NSE and RSR as suggested above will likely show that the performance is not very good.

---

## Referee Comment (RC3) · Anonymous Referee #3 · 26 Jan 2020

This manuscript describes three approaches (ANN, Exponential filter, and CDF) to estimate subsurface soil moisture from surface soil moisture data in the Qilian Mountains of China. Authors identified the Exponential filter as the best model and applied this model in different ways throughout the manuscript. The topic is of great interest, but I think that the manuscript requires a significant restructuring in order to be considered acceptable for publication on HESS. My major concerns are:

1) The organization of the manuscript and its presentation is not fluent. It seems that a series of tests and analysis have been listed one after the other without a logic.

2) For instance, I do not see any added value is the preliminary analysis of the soil moisture data. It is quite obvious that surface and subsurface soil moisture are linked or coupled. Remove this part or avoid stating that it is an outcome of the study.

[Figure]

3) Second step in the manuscript is the intercomparison of different models. In this step, it seems that the use of ANN is made just applying a matlab tool without providing enough details about the approach adopted.

4) The intercomparison may be influenced by the different approaches used for the calibration of the methods. In fact, authors states that 70% of the data was used for validation of ANN and CDF, but they do not provide such indication for the exponential filter. If they used the entire database for this last, this may affect the results.

5) I personally do not understand the need to include a section of the cross-correlation analysis. It seems out of the scope of the manuscript. Moreover, no significant result are discussed herein. Please remove this section.

6) Authors proposed some multilinear functions to describe relative value of T, which is fine, but it is not connected with anything else in the manuscript. It is another element somewhat independent from the main objective of the manuscript. Consider to

7) In the last section, we start another new section where SMAP is used first in comparison with the observation revealing some limitation for higher values. Such statement should take into consideration the existing gap in the spatial resolution of the two measurements. Rough resolution tend to smooth out higher values. This is quite obvious.

8) Finally, authors close with a comparison of exponential filter applied on SMAP. The regression are not used for this scope, other methods are not considered in this section, crosscorrelation and spatial dynamics also neglected. I reached this point and I realised that authors are following a random walk of activities and I felt confused and disoriented.

This manuscript requires a DRASTIC RESTRUCTURING and REORGANIZAITON before being considered for publication. It will also benefit of a significant shortening of useless contents.
603, 2019.

---

## Author Comment (AC2) · 23 Feb 2020

The author's answers are indicated in red color, as well as old text passages. New text passages are indicated in green color.

1. The aim of this study is ambiguous. Is it comparison of different methods, improvement of methods, or evaluation of satellite products?

Response:

The aim of this study is to use multi-station in situ SM observations and remotely sensed SM data from the Qilian Mountains, a prime example of a high and cold mountainous area, to characterize the relationship between surface SM and deeper SM in order to obtain the spatial distribution of profile SM.

In the revised manuscript, we have deleted some contents that are not important for the analysis, and we made a drastic restructuring and reorganization to make the revised manuscript easier to understand.

The revised manuscript is now divided into three parts. Firstly, we evaluated the different methods for estimating subsurface soil moisture (SM). The ExpF method was found to be the most suitable method for further application in the study area.

Secondly, our results indicate that the median value of $T_{opt}$ can be used for application of the ExpF method in the study area.

Finally, the ExpF method derived with the median value of $T_{opt}$ was combined with the SMAP_L3 surface SM product to estimate the subsurface SM. The subsurface SM was also compared to the SMAP_L4 root zone SM product (a widely used large-scale root zone SM product). Results indicated that the combination of the ExpF method with the SMAP_L3 surface SM product can significantly improve the estimation of profile SM in mountainous areas. Furthermore, the combination of SMAP_L3 and the ExpF method (with the median value of Topt) was applied to estimate the temporal and spatial distribution of profile SM in the study area.

2. Does the ExpF method with optimum Topt perform better than the ANN? If not, why does the author apply the ExpF method to expand SMAP?

Response:

Thank you for your comment and suggestion. The ANN method is statistically superior to those from the other two methods. However, we prefer the ExpF approach as it is simpler to apply and more process-based than the ANN method. in the revised manuscript. (Line 213-223)

As expected, all metrics showed that the performance decreased with depth. The results indicate that for two out of the three statistical measures (RSR and NSE), the ANN method was statistically superior to the other two methods. Specifically, the ANN method resulted in the lowest estimation error, while the ExpF method was better able to capture SM dynamics. A similar finding was reported by Zhang et al. (2017a), who found that the ExpF method had a significantly higher correlation coefficient along with a higher mean bias compared to the ANN method. Results also demonstrated that both ANN and ExpF method were able to provide accurate estimates of subsurface SM for layer 2, layer 3 and profile SM.

Overall, both the ANN and ExpF method are useful for estimating subsurface SM from surface SM in our study area, and ANN was the statistically superior method. However, the ExpF method is a simpler approach as it only needs one parameter ($T_{opt}$), and can thus be easily applied in data-scarce mountainous areas. Therefore, the ExpF method was used to estimate subsurface SM in the remainder of this study.

3. In the introduction, the author mentions there are four groups of methods, what are their advantages and disadvantages? Why did the author choose the three methods in this study?

Response:

In the introduction, we introduced five groups of methods, which are data assimilation of remote sensing data into land surface model (Han et al., 2013), physically-based methods (Manfreda et al., 2014), (semi-) empirical approaches (Albergel et al., 2008), data-driven methods (Kornelsen and Coulibaly, 2014; Zhang et al., 2017a), and statistical methods (Gao et al., 2019). Among them, the application of both data assimilation and physically based methods are limited due to the large amount of required input data, e.g. soil properties, which are often not available for data-scarce mountainous areas (Jin et al., 2015; Li et al., 2017; Dai et al., 2019). As we want to evaluate the methods that can be used in data-scarce mountainous areas, we exclude both data assimilation and physically based methods, and evaluated the other three methods in this study.

4. In section 4.1, there are lag time between soil moisture data at different layers and at surface. How did the author consider the impacts of the lag time in applications of these methods?'

Response:

Thanks for your suggestion. We wanted to evaluate the coupling strength between surface soil moisture and subsurface soil moisture. However, there is no satisfying criterion to conclude whether the coupling strength is strong or not according to the results of the cross-correlation analysis. Thus, we have deleted the contents related to the cross-correlation analysis in the revised manuscript.

5. The performance of the ANN method is significantly related to the training data. In this study, 70 % data was used as training data according to Zhang's study. However, Zhang's study focused on the US., is 70 % suitable for the high mountainous area? Moreover, even with a ratio of 70%, there are lots of data combinations, what's the principle to choose these data? Does the author compare the performance of the ANN method with different data combinations?

Response:

Firstly, for the ANN method, the sample number of training has no relation with the location, and 70% is usually selected as the number of the training samples (e.g. Kornelsen and Coulibaly, 2014; Zhang et al., 2017). The training with 70% of the data was also used in the estimation of soil moisture time series from passive microwave data using the ANN method in the Heihe River watershed (Lu et al., 2017).

Secondly, we used random sampling with uniform distribution in this study, which can best balance the induced error by data sampling. Then, the combination with the best metric (minimum RMSE) was selected for the ANN method.

6. In section 4.3.2, Topt is estimated by precipitation and clay ratio. However, the main advantage of the RBF method is its requirement of few data in introduction. Thus, the improvement in this study is meaningless. What's the insight of this improvement in other regions?

Response:

The correlation between ln-transformed LAI and precipitation is significant (Pearson's R=0.80, P<0.01). Furthermore, we tested the partial correlation analysis of the ln-transformed LAI, precipitation and Topt. The results showed that the relationships between ln-transformed LAI and $T_{opt}$ are nonsignificant under the control of precipitation. Meanwhile, the relationships between

precipitation and $T_{opt}$ under the control of ln-transformed LAI are not valid for all layers. Thus, this section about the control factors of $T_{opt}$ is not convincing. Furthermore, as the control factors and regression of Topt are not applied to the further estimation of subsurface soil moisture from the SMAP_L3 product, this part is not important for the manuscript. Therefore, we have deleted the section on the control factors and regression of $T_{opt}$ in the revised manuscript.

We improved the estimation of profile soil moisture in data-scarce mountainous areas as follows. Our study evaluates four methods to estimate $T_{opt}$ in the data-scarce Qilian mountains, and the results show that the area-specific estimation of $T_{opt}$ (site-specific $T_{opt}$, area-generalized $T_{opt}$) has significantly higher performance than the widely-used $T_{opt}$ ($T_{Warger}$ and $T_{Qiu}$), which has been applied in cold mountainous areas (e.g. the utility of $T_{opt}$=20 days for profile SM estimation in east Asia in Muhammad et al. (2017)).

Furthermore, the results indicate that there is a non-significant difference between the performance of a site-specific $T_{opt}$ and an area-generalized $T_{opt}$. Thus, the area-specific Topt can be combined with ExpF method to estimate profile soil moisture with good performance. The reference $T_{opt}$ for the estimation of profile and subsurface soil moisture in the study area are provided in the manuscript, and provide a reference for future studies in similar areas.

Finally, we compared the estimation of profile soil moisture based on the combination of the SMAP_L3 surface product and the ExpF method (with a median value of $T_{opt}$ of SMAP) with a widely-used profile soil moisture product (SMAP_L4 root zone product). Results showed that our method can improve the profile soil moisture estimation significantly in our study area. Thus, based on the large-scale in-situ observations, we believe that our study improves the estimation of profile soil moisture in cold mountain areas, which will be useful for water resources management in inland river basins.

7. The author evaluates both SMAP_L3 and SMAP_L4 products against in situ observations. The SMAP_L4 is the assimilation results of satellite data and model simulation. What's the impacts of their original biases of SMAP_L3 and SMAP_L4 respectively? What's the impacts of scale-mismatch between footprint scale of satellite products and point scale of in situ observations?

Response:
In this study, we have we partitioned the bias in the SMAP-based estimation of profile SWI

("SMAP-observed profile SWI", Fig. S8(c)) in bias associated with the ExpF method and bias due to SMAP original differences to get insight into the major source of error in SMAP-based estimates of profile SWI. Results showed that the major error stems from the SMAP_L3 product.

The SMAP_L4 product is widely-used large-scale root zone soil moisture product. It is used as reference to test whether our method can improve profile soil moisture estimation. The results indicate that the ExpF method combined with the SMAP_L3 can significantly improve profile soil moisture estimation.

We have noted the problem of scale mismatch between the in-situ observations and the SMAP product. We have added a discussion about the introduced error from the scale mismatch in the revised manuscript (Line 297-300).

Notably, the SMAP_L3 product has a spatial posting of 9 km×9 km, while the in-situ measurements are point-based and soil moisture has a strong spatial variability in mountainous areas (Tian et al., 2019). Thus, the disparity of spatial scales between points and satellite footprints will introduce additional errors in the validation of the satellite products (Jin et al. 2017).

**Reference:**

Albergel C, Rüdiger C, Pellarin T et al. From near-surface to root-zone soil moisture using an exponential filter: an assessment of the method based on in-situ observations and model simulations. Hydrology and Earth System Sciences, 2008, 12(6): 1323-1337.

Dai Y, Shangguan W, Wei N et al. A review of the global soil property maps for Earth system models. SOIL, 2019, 5(2): 137-158.

Gao X, Zhao X, Brocca L, Pan D, Wu P. Testing of observation operators designed to estimate profile soil moisture from surface measurements. Hydrological Processes, 2019.

Han X, Hendricks Franssen H-J, Li X, Zhang Y, Montzka C, Vereecken H. Joint assimilation of surface temperature and L-band microwave brightness temperature in land data assimilation. Vadose Zone Journal, 2013, 12(3).

Jin R, Li X, Liu S M. Understanding the Heterogeneity of Soil Moisture and Evapotranspiration Using Multiscale Observations From Satellites, Airborne Sensors, and a Ground-Based Observation Matrix. IEEE Geoscience and Remote Sensing Letters, 2017, 14(11): 2132-2136.

Jin X, Zhang L h, Gu J, Zhao C, Tian J, He C S. Modeling the impacts of spatial heterogeneity in soil hydraulic properties on hydrological process in the upper reach of the Heihe River in the

Qilian Mountains, Northwest China. Hydrological Processes, 2015, 29(15): 3318-3327.

Li X, Liu S, Xiao Q et al. A multiscale dataset for understanding complex eco-hydrological processes in a heterogeneous oasis system. Scientific data, 2017, 4: 170083.

Lu Z, Chai L, Liu S, Cui H, Zhang Y, Jiang L, Jin R, Xu Z. Estimating Time Series Soil Moisture by Applying Recurrent Nonlinear Autoregressive Neural Networks to Passive Microwave Data over the Heihe River Basin, China. Remote Sensing, 2017, 9(6): 574.

Manfreda S, Brocca L, Moramarco T, Melone F, Sheffield J. A physically based approach for the estimation of root-zone soil moisture from surface measurements. Hydrology and Earth System Sciences, 2014, 18(3): 1199-1212.

Muhammad Z, Hyunglok K, Minha C. Evaluating the patterns of spatiotemporal trends of root zone soil moisture in major climate regions in East Asia. Journal of Geophysical Research: Atmospheres, 2017, 122(15): 7705-7722.

Tian J, Zhang B, He C, Han Z, Bogena H R, Huisman J A. Dynamic response patterns of profile soil moisture wetting events under different land covers in the Mountainous area of the Heihe River Watershed, Northwest China. Agricultural and Forest Meteorology, 2019, 271(15): 225-239.

Zhang N, Quiring S, Ochsner T, Ford T. Comparison of Three Methods for Vertical Extrapolation of Soil Moisture in Oklahoma. Vadose Zone Journal, 2017, 16(10).

---

## Author Comment (AC3) · 23 Feb 2020

The author's answers are indicated in red color, as well as old text passages. New text passages are indicated in green color.

General comments: This manuscript describes a wide-variety of approaches to estimate subsurface and profile soil moisture from surface soil moisture data in the Qilian Mountains of China. Most of the conclusions are well-supported, but the manuscript suffers from statistical inconsistencies. A consistent set of statistical measures should be maintained throughout the manuscript (NSE, RSR, and R). The only exception could be Fig. 10 where the aim is to compare the SMAP ubRMSE to the mission's accuracy requirement. Also, 30% of the data should be withheld for validation for all three methods, not only the ANN method. The performance of the SMAP-based ExpF method for estimating profile soil moisture (i.e., SWI) is overstated, and including the NSE and RSR statistics will likely provide a much more objective view. The manuscript also suffers from poor organization in some places and is not well-written.

Response:
Thanks for your comments.

We have maintained the consistency of the statistics throughout the manuscript, expect for the use of ubRMSE in the evaluation of SMAP product.

The $T_{opt}$ parameter of the ExpF method reflects the characteristic length of the temporal dynamics of soil moisture. Earlier studies revealed that $T_{opt}$ is highly dependent on the sampling interval of soil moisture data (De Lange et al., 2008). In our study, we found that when using the random sampling with 70% training data as for the ANN method, $T_{opt}$ was not suitable for the remaining data. Since it was not possible to use the same training method for ExpF method as for ANN, we used the entire soil moisture time series to estimate $T_{opt}$, which was also the standard procedure in earlier studies (e.g. Wagner et al., 1999; Albergel et al., 2008; De Lange et al., 2008; Ford et al., 2014; Wang et al., 2017).

In the revised manuscript, the performance of SMAP-based ExpF method for estimating profile soil moisture has been reanalyzed using the metrics of NSE and RSR.

In addition, we have deleted some content that is not important for the analysis and we have reorganized the text to make the revised manuscript easier to understand.

The revised manuscript is now divided into three parts. Firstly, we evaluated the different methods for estimating subsurface soil moisture (SM). The ExpF method was found to be the most suitable method for the further application in the study area.

Secondly, our results indicate that the median value of $T_{opt}$ can be used for application of the ExpF method in the study area.

Finally, the ExpF method derived with the median value of $T_{opt}$ was combined with the SMAP_L3

surface SM product to estimate the subsurface SM. The subsurface SM was also compared to the SMAP_L4 root zone SM product (a widely used large-scale root zone SM product). Results indicated that the combination of the ExpF method with the SMAP_L3 surface SM product can significantly improve the estimation of profile SM in mountainous areas. Furthermore, the combination of the SMAP_L3 and ExpF method (with the median value of Topt) were applied to estimate the temporal and spatial distribution of profile SM in the study area.

Specific comments:

1. line 126. Clarify the meaning of "to consider the assumption of uniform vertical profiles of soil temperature and soil dielectric properties".

Response:

We have deleted it in the revised manuscript:

2. Table 1. Results are being reported with too many significant figures. I doubt that the laboratory is able to measure sand and silt content with adequate precision to justify four significant figures. Reduce to a more appropriate level, perhaps three significant figures.

Response:

We have reduced the level to three significant figures in the revised manuscript:

**Table 1.** Statistics of the soil physical characteristics at the 35 soil moisture stations: mean (standard deviation)

| Layer | Depth (cm) | Bulk Density (g/cm$^3$) | $K_S$ (cm/hour) | SOC (g/100g) | Sand (%) | Silt (%) | Clay (%) |
|---|---|---|---|---|---|---|---|
| Layer 1 | 0-10 | 1.13(0.28) | 3.87(4.11) | 4.35(4.11) | 26.6(11.9) | 66.2(10.9) | 7.2(1.6) |
| Layer 2 | 10-20 | 1.14(0.24) | 4.61(4.53) | 3.9(3.87) | 24.5(11.9) | 68.6(11.2) | 6.9(1.2) |
| Layer 3 | 20-30 | 1.18(0.32) | 4.78(6.22) | 3.63(3.54) | 27.0(15.2) | 66.5(14.3) | 6.5(1.4) |
| Layer 4 | 30-50 | 1.29(0.3) | 3.94(4.68) | 2.21(2.28) | 29.5(15.3) | 63.8(14.5) | 6.5(1.6) |
| Layer 5 | 50-70 | 1.34(0.3) | 1.85(2.35) | 2.34(2.47) | 26.9(17.1) | 66.5(15.9) | 6.7(1.9) |

Note: $K_S$ is the Saturated Hydraulic Conductivity; SOC is the Soil Organic Carbon.

3. Section 3.4. Provide more explanation. Was a separate ANN model developed for every depth combination and every site?

Response:

Yes. In this study, a separate ANN model was developed for every depth of every site. We have added more details about the set-up of the ANN method in the revised manuscript (Section 3.3, Line 153-165).

4. Equation 6 is the wrong equation.

Response:

We have corrected Equation 6 in the revised manuscript. (Line 173)

$$\hat{\Delta} = K_0 + K_1 \, \theta_1 + K_2 \cdot \theta_1{}^2 + K_3 \cdot \theta_1{}^3 \qquad\qquad (9)$$

Where $\hat{\Delta}$ is the predicted difference between surface and subsurface SM, and $K_i$ (i=0,1,2,3) are parameters.

5. line 196. not "persistent" but "consistent"

Response:

As suggested by referee 3, we have deleted this part.

6. line 205-208. This should be moved to the methods section.

Response:

As suggested by referee 3, we have deleted this part.

7. line 257. You have not provided any convincing evidence that "For most hydrological researches, the correct temporal variation of SM is more crucial than the exact value, suggesting that more emphasis should be given to R when selecting the most appropriate estimation method." You have presented three statistical measures to evaluate these methods (RSR, R, and NSE). For two out of the three statistical measures (RSR and NSE) the ANN method had the best performance. Therefore, you should include a clear statement that the results from the ANN method were statistically superior to those from the other two methods. You are still free to prefer the ExpF approach if it is simpler to apply than the ANN method. Just don't try to justify that choice on a statistical basis.

Response:

Thank you for your comment and suggestion. Yes, the ANN method is statistically superior to those from the other two methods. However, we prefer the ExpF approach as it is simpler to apply and more process-based than the ANN method. We have deleted the statement "For most hydrological researches, the correct temporal variation of SM is more crucial than the exact value, suggesting that more emphasis should be given to R when selecting the most appropriate estimation method" in the revised manuscript. Furthermore, we have included the statement "The results suggested that for two out of the three statistical measures (RSR and NSE), the ANN method was statistically superior to those from the other two methods." in the revised manuscript. (Line 213-223)

As expected, all metrics showed that the performance decreased with depth. The results indicate that for two out of the three statistical measures (RSR and NSE), the ANN method was statistically superior to the other two methods. Specifically, the ANN method resulted in the lowest estimation error, while the ExpF method was better able to capture SM dynamics. A similar finding was

reported by Zhang et al. (2017a), who found that the ExpF method had a significantly higher correlation coefficient along with a higher mean bias compared to the ANN method. Results also demonstrated that both ANN and ExpF method were able to provide accurate estimates of subsurface SM for layer 2, layer 3 and profile SM.

Overall, both the ANN and ExpF method are useful for estimating subsurface SM from surface SM in our study area, and ANN was the statistically superior method. However, the ExpF method is a simpler approach as it only needs one parameter ($T_{opt}$), and can thus be easily applied in data-scarce mountainous areas. Therefore, the ExpF method was used to estimate subsurface SM in the remainder of this study.

8. Figure 7a is unnecessary and should be deleted. Your results show that "Year" does not have a significant effect, so the data should be presented including all years as done in Fig. 7b.

Response:
Thank you for your comment and suggestion. We have now merged Figure 5 into Figure 4 in the revised manuscript (Fig. 4).

[Figure]

Fig. 4. Variation of NSE with T of the exponential filter method at different layers of each stations during the growing season of 2014, 2015 and 2016. Y axis is the NSE value. Frequency distribution curve and the boxplots to show the distribution of $T_{opt}$ with depth for all stations.

9. line 290-297. This should be moved to the methods section.

Response:

Thank you for your comment and suggestion. As suggested in Comment 10, the correlation among variables (e.g. Ln-transformed LAI and precipitation) are strong, and the results of partial correlation analysis among $T_{opt}$ and the variables indicated that the relationship among $T_{opt}$ and variables are not valid. Therefore, the relationships between $T_{opt}$ and variables are not applied in the further analysis. As a result, this part has been deleted in the revised manuscript.

10. line 298-305. This section is not convincing. How strong is the correlation between ln-transformed LAI and precipitation? Perhaps the apparent relationship between Topt and LAI is a spurious result of the correlation between LAI and precipitation.

Response:

Thank you for your comment and suggestion. The correlation between ln-transformed LAI and

precipitation are significant (Pearson's R=0.80, P<0.01). Furthermore, we tested the partial correlation analysis of the ln-transformed LAI, precipitation and $T_{opt}$. The results showed that the relationships between ln-transformed LAI and $T_{opt}$ are non-significant under the control of precipitation. Meanwhile, the relationships between precipitation and $T_{opt}$ under the control of ln-transformed LAI are not valid for all layers. Thus, this section on the control factors of $T_{opt}$ is not convincing and was removed.

Furthermore, as the control factors and regression of $T_{opt}$ are not applied any more to the further estimation of subsurface soil moisture from the SMAP_L3 product. Thus, we have deleted the content on the control factors and regression of $T_{opt}$ in the revised manuscript.

11. line 319-322. Move to methods.

Response:

We have moved it to the Methods section (Section 3.3).

12. line 319-322. What steps were taken to prevent problems due to collinearity of the predictor variables?

Response:

Firstly, the regression equation was not used in the further analysis. Secondly, it is difficult to obtain independent variables for the regression equations in data-scarce mountainous areas. Thus, the regression equation for $T_{opt}$ should not be used in such environments. Therefore, we have deleted this part in the revised manuscript.

13. Table 4. Is "ln ln (sand)" correct in the last row?

Response:

As stated in the reply for comment 10, this part has been deleted in the revised manuscript.

14. Fig. 9. Present RSR instead of RMSE to be consistent with the rest of the manuscript.

Response:

We have changed the RMSE with RSR in the revised manuscript. (Fig. 5)

[Figure]

**Fig. 5.** The boxplot of NSE, Pearson's R, and RMSE for the $T_{opt}$ generated from different schemes. The different letters above box indicate the significant difference for different schemes.

15. line 380. Not "persistent" but "consistent".

Response:
We have changed it. (Line 283)

16. Section 4.4.1. You should note an important limitation of this analysis. There is a huge scale mismatch between the 9 km SMAP data and the in situ sensors which measure at a single point. This will likely degrade the agreement between the two data sets.

Response:
Due to the harsh natural environment of the Qilian Mountain, there is still a lack of high-precision soil property data in the study area (Li et al. 2017; Tian et al. 2017; Li et al., 2018; Zhao et al. 2018), which is necessary to downscale SMAP soil moisture data (Montzka et al. 2018). In general, large research gaps still exist with respect to scaling issue in mountainous areas with complex topography (Jin et al. 2017; Fan et al. 2019). Such scale issues are beyond the scope of this manuscript. We have added a discussion about the limitations in the manuscript. (Line 297- 300)

Notably, the SMAP_L3 product has a spatial posting of 9 km×9 km, while the in-situ measurements are point-based and soil moisture has a strong spatial variability in mountainous areas (Tian et al., 2019). Thus, the disparity of spatial scales between points and satellite footprints will introduce additional errors in the validation of the satellite products (Jin et al. 2017).

17. line 394-399. Move this to methods.

Response:
We have moved it to Data and Methods section (Section 3.1). (Line 122-124)

SMAP_L3 surface soil moisture was also used to estimate subsurface soil moisture (Layer 2: 10-20 cm, Layer 3: 20-30 cm, Layer 4: 30-50 cm, Layer 5: 50-70 cm, Profile: 0-70 cm) during the growing seasons of 2015 and 2016 in the study area.

18. line 394-399. Why did you even bother all the effort to determine Topt from the in situ stations in the prior sections? Now you are not using those Topt values but instead finding new ones based on comparison of the SMAP data with the in situ data. This does not make sense in the flow of the manuscript.

Response:

Firstly, the ExpF method only need one parameter ($T_{opt}$) to estimate the subsurface soil moisture. In order to use the exponential filter method in areas with limited or no soil moisture observations, especially in high and cold mountainous areas, it is necessary to understand local controls on the estimation of $T_{opt}$. Therefore, we put more effort on analyzing $T_{opt}$.

Secondly, the $T_{opt}$ from in-situ stations were used to test the alternative methods for estimation of $T_{opt}$ in the study area. This analysis showed that the median value of $T_{opt}$ is robust in the study area.

Finally, earlier studies revealed that $T_{opt}$ highly depended on the sampling interval of soil moisture data (De Lange et al., 2008). As data with the longer time interval typically have higher $T_{opt}$ values, $T_{opt}$ of in-situ data are mostly lower than satellite retrievals (Albergel et al. 2008; Ford et al., 2014; González-Zamora et al., 2016). In this study, the in-situ soil moisture observations are daily scale (the mean and standard deviation of the time interval for the 35 sites are 1 days and 0.3 days, respectively). However, as SM estimation from spaceborne sensors is still challenging for the high and cold mountainous regions, data gaps exist for the SMAP_L3 data in our study area (the mean and standard deviation of the time interval for the 35 sites are 2.3 days and 1.6 days, respectively). Thus, the time interval of the SMAP_L3 data is larger than the in-situ observations. Thus, the $T_{opt}$ of SMAP is different from the $T_{opt}$ of in-situ observations. Consequently, to get the best estimation of subsurface soil moisture at large scale, the subsurface SWI was estimated through combination of SMAP surface soil moisture with the ExpF method (with the site-specific $T_{opt}$ based on the best match between observations in terms of NSE).

19. line 400-406. Include NSE and RSR measures here. They are crucial for quantifying the mismatch between the SMAP SWI and the observed SWI values as shown in Fig. 11.

Response:
We have included the NSE and RSR here. (Line 310-317)

As expected, the estimation accuracy of subsurface SM decreased with depth. The ANOVA results showed that the subsurface SM estimation accuracy for layer 2 (median

value of RSR=0.92, R=0.69, NSE=0.18) and profile SM (RSR=0.92, R=0.65, NSE=0.14) were significantly higher than for layer 4 (RSR=1.12, R=0.31, NSE=-0.13) and layer 5 (RSR=1.17, R=0.34, NSE=-0.15) ($p<0.05$). The NSE values were positive for layer 2 and profile SM, while the NSE values for the other layers were negative. The negative MBE showed that subsurface SM was underestimated. The relationship between SMAP-derived and in-situ observed subsurface SM for layer 2 and profile SM was significant ($p<0.01$) at all but one station (D15). Thus, the SMAP surface product and ExpF method can be used to estimate the subsurface SM in the study area, especially for layer 2 (10-20 cm) and profile (0-70 cm) SM.

20. line 425-428. This point should also have been made in Section 4.4.1.

Response:

We have added the error caused by the scale mismatch between point measurements and satellite footprints in Section 4.3.1 of the revised manuscript. (Line 291-294)

Here, it is important to note that the SMAP_L3 product is provided at a 9 km × 9 km resolution while the in-situ measurements are point-based and soil moisture has a strong spatial variability in mountainous areas (Tian et al., 2019). Thus, part of the variability in Fig. 6 is due the disparity of spatial scales between the point-scale and the satellite footprint (Jin et al. 2017).

21. Tables 5 and 6. Replace RMSE with RSR. Add NSE.

Response:

We have replaced the RMSE with RSR and added the NSE in Tables 4 and 5. The analysis of the results was also changed according to the metrics of RSR, R and NSE in the revised manuscript. (Table 4 and Table 5, as the original Table 4 is the results of regression equation for $T_{opt}$, it has been deleted in the revised manuscript)

**Table 4.** Statistics of the metrics (RSR, R, NSE) of the comparisons of SMAP estimated and observed SWI at different layers for the 35 stations during the growing seasons of 2015-2016.

| Layer | RSR | | R | | NSE | |
|---|---|---|---|---|---|---|
| | Mean±Std | Median | Mean±Std | Median | Mean±Std | Median |
| Layer 2 a | 1.21±1.27 | 0.90 | 0.64±0.24 | 0.73a | 0.04±0.49 | 0.19 |
| Layer 3 ab | 1.27±0.93 | 1.06 | 0.55±0.31 | 0.64ab | -0.05±0.46 | 0.05 |

| | | | | | | |
|---|---|---|---|---|---|---|
| Layer 4 b | 1.53±1.49 | 1.10 | 0.43±0.38 | 0.52b | -0.22±0.53 | -0.08 |
| Layer 5 b | 2.03±3.48 | 1.16 | 0.41±0.39 | 0.55b | -0.27±0.61 | -0.14 |
| Profile a | 1.18±0.74 | 0.88 | 0.63±0.27 | 0.72a | 0.12±0.38 | 0.19 |

Note: the different letters after the layers indicate that the difference is significant at p<0.05 (Kruskal-Wallis ANOVA)

**Table 5.** Statistics of the metrics (RSR, R, NSE) of the comparisons of estimated-observed profile SWI datasets, SMAP_L3-observed surface SWI datasets, SMAP_L3-observed profile SWI datasets, and SMAP_L4-observed profile SWI datasets for the 35 stations during the growing season of 2015-2016.

| Comparisons | RSR | | R | | NSE | |
|---|---|---|---|---|---|---|
| | Mean±std | Median | Mean±std | Median | Mean±std | Median |
| Estimated-observed PSWI | 0.86±1.00 | 0.68 | 0.88±0.11 | 0.9 | 0.56±0.32 | 0.64 |
| SMAP_L3-observed SSWI | 1.13±0.49 | 1.01 | 0.57±0.17 | 0.59 | -0.09±0.52 | -0.07 |
| SMAP_L3-observed PSWI | 1.18±0.74 | 0.88 | 0.63±0.27 | 0.72 | 0.12±0.38 | 0.19 |
| SMAP_L4-observed PSWI | 1.42±0.76 | 1.25 | 0.47±0.31 | 0.55 | -0.49±0.68 | -0.3 |

Note: e.g. Estimated-observed PSWI means the comparison of the estimated profile SWI and observed profile SWI.

Fig. S8 and Table 5 show that the SMAP-observed SWI comparisons had lower performance metrics for surface SWI (median value of RSR, R and NSE are 1.01, 0.59 and -0.07, respectively) than for profile SWI (median value of RSR, R and NSE are 0.88, 0.72 and 0.19, respectively). A possible reason for this is that the profile SWI was estimated based on the SMAP surface SWI and $T_{opt}$. The latter was determined by optimization using the maximum NSE, which improved the performance of profile SWI estimation. In addition, the performance metrics for SMAP–observed SWI comparisons for both surface and profile SWI were significantly (p<0.001) lower than those of estimated–observed profile SWI (median value of RSR, R and NSE are 0.68, 0.90 and 0.64, respectively). Thus, the major error in SMAP-based profile SWI estimates stems from the SMAP satellite product and is not derived from ExpF method, which is also supported by other researches (e.g. Ford et al., 2014; Pablos et al., 2018). Notably, the scale mismatch between point measurements and satellite footprints will introduce additional errors in the validation of the satellite estimated subsurface products (Jin et al. 2017).

22. line 436. Your results (Fig. 11) show that the performance of the SMAP profile SWI estimates is relatively poor. This is being partly obscured by the omission of the NSE and RSR statistics.

Response:

Thank you for your comments and suggests. We have added NSE and RSR in the revised manuscript. As shown in Tables 4 and 5, the SMAP estimated profile SWI has a median value of RSR=0.88, R=0.72 and NSE=0.19, respectively. Although the NSE is low, its performance is significantly

(p<0.001) higher than the SMAP_L4 profile SWI in our study area.

23. line 451-452. Here on the 22nd page of the manuscript a completely new data set is introduced. This is inappropriate. If this section is important to the manuscript, then take the time to justify it in the introduction and describe it in the methods.

Response:
We have deleted this part in the revised manuscript.

24. line 465. Interpolated how? What evidence do you have that the interpolation is statistically valid? What is the associated uncertainty? This again should be justified in the introduction and described in the methods.

Response:
As suggested in comment 25, we can use the median value of $T_{opt}$ instead of using the site-specific $T_{opt}$ value. Thus, the median value of $T_{opt}$ ($T_{opt}$=10 days) was used to derive ExpF method in estimating the profile SWI in the study area in the revised manuscript. We have deleted the content related to the interpolation of $T_{opt}$ and the Fig. S14 (spatial distribution of $T_{opt}$).

25. line 465. Also, why bother to spatially interpolate Topt? You have just argued that Topt defined in one region (Heihe) is valid in another region (Maqu).

Response:
As argued in Section 4.3.3, the median value of $T_{opt}$ can be used to derive the ExpF method for estimating profile soil moisture. For the estimation of profile SWI from SMAP surface product, we used the median value of $T_{opt}$ ($T_{opt}$=10 days) instead of the interpolated $T_{opt}$ to derive the ExpF method in the revised manuscript.

26. line 495. The data in Fig. 11 show that the accuracy is relatively poor. Relying on the R value alone is clearly misleading in this case where there is a substantial bias. Including the NSE and RSR as suggested above will likely show that the performance is not very good.

Response:
We have included NSE and RSR in the analysis (Tables 4 and 5 in the revised manuscript). This is now mentioned in the revised manuscript accordingly. (Line 382-384)

   4) The ExpF method is useful and potential for estimating profile SM from SMAP_L3 surface product. As it can significantly improve the estimation of profile SM in cold arid mountainous areas (e.g. compared to the SMAP_L4 root zone product), and the main error stems from the satellite product.

**Reference:**

Albergel C, Rüdiger C, Pellarin T et al. From near-surface to root-zone soil moisture using an exponential filter: an assessment of the method based on in-situ observations and model simulations. Hydrology and Earth System Sciences, 2008, 12(6): 1323-1337.

Fan L, Al-Yaari A, Frappart F et al. Mapping Soil Moisture at a High Resolution over Mountainous Regions by Integrating In Situ Measurements, Topography Data, and MODIS Land Surface Temperatures. Remote Sensing, 2019, 11(6): 656.

Ford T W, Harris E, Quiring S M. Estimating root zone soil moisture using near-surface observations from SMOS. Hydrology and Earth System Sciences, 2014, 18(1): 139-154.

González-Zamora Á, Sánchez N, Martínez-Fernández J, Wagner W. Root-zone plant available water estimation using the SMOS-derived soil water index. Advances in Water Resources, 2016, 96: 339-353.

Jin R, Li X, Liu S M. Understanding the Heterogeneity of Soil Moisture and Evapotranspiration Using Multiscale Observations From Satellites, Airborne Sensors, and a Ground-Based Observation Matrix. IEEE Geoscience and Remote Sensing Letters, 2017, 14(11): 2132-2136.

Lange R d, Beck R, Giesen N v d, Friesen J, Wit A d, Wagner W. Scatterometer-Derived Soil Moisture Calibrated for Soil Texture With a One-Dimensional Water-Flow Model. IEEE transactions on Geoscience and remote sensing, 2008, 46(12): 4041-4049.

Li J, Zhang L, He C, Zhao C. A Comparison of Markov Chain Random Field and Ordinary Kriging Methods for Calculating Soil Texture in a Mountainous Watershed, Northwest China. Sustainability, 2018, 10(8): 2819.

Li X, Liu S, Xiao Q et al. A multiscale dataset for understanding complex eco-hydrological processes in a heterogeneous oasis system. Scientific data, 2017, 4: 170083.

Montzka C, Rötzer K, Bogena H R, Sanchez N, Vereecken H. A New Soil Moisture Downscaling Approach for SMAP, SMOS, and ASCAT by Predicting Sub-Grid Variability. Remote Sensing, 2018, 10(3): 427.

Pablos M, González-Zamora Á, Sánchez N, Martínez-Fernández J. Assessment of Root Zone Soil Moisture Estimations from SMAP, SMOS and MODIS Observations. Remote Sensing, 2018, 10(7): 981.

Tian J, Zhang B, He C, Han Z, Bogena H R, Huisman J A. Dynamic response patterns of profile soil moisture wetting events under different land covers in the Mountainous area of the Heihe River Watershed, Northwest China. Agricultural and Forest Meteorology, 2019, 271(15): 225-239.

Tian J, Zhang B, He C, Yang L. Variability in Soil Hydraulic Conductivity and Soil Hydrological Response Under Different Land Covers in the Mountainous Area of the Heihe River Watershed, Northwest China. Land Degradation & Development, 2017, 28(4): 1437-1449.

Wagner W, Lemoine G, Rott H. A Method for Estimating Soil Moisture from ERS Scatterometer and Soil Data. Remote Sensing of Environment, 1999, 70(2): 191-207.

Wang T, Franz T E, You J, Shulski M D, Ray C. Evaluating controls of soil properties and climatic conditions on the use of an exponential filter for converting near surface to root zone soil moisture contents. Journal of Hydrology, 2017, 548: 683-696.

---

## Author Comment (AC4) · 23 Feb 2020

The author's answers are indicated in red color, as well as old text passages. New text passages are indicated in green color.

This manuscript describes three approaches (ANN, Exponential filter, and CDF) to estimate subsurface soil moisture from surface soil moisture data in the Qilian Mountains of China. Authors identified the Exponential filter as the best model and applied this model in different ways throughout the manuscript. The topic is of great interest, but I think that the manuscript requires a significant restructuring in order to be considered acceptable for publication on HESS. My major concerns are:

Response:
Thanks for your comments. We have made a significant restructuring in the revised manuscript.

1. The organization of the manuscript and its presentation is not fluent. It seems that a series of tests and analysis have been listed one after the other without a logic.

Response:
Thanks for your comments. In the revised manuscript, we have deleted some contents that are not important for the analysis, and we made a drastic restructuring and reorganization to make the revised manuscript easier to understand.

The revised manuscript is now divided into three parts. Firstly, we evaluated the different methods for estimating subsurface soil moisture (SM). The ExpF method was found to be the most suitable method for further application in the study area.

Secondly, our results indicate that the median value of $T_{opt}$ can be used for application of the ExpF method in the study area.

Finally, the ExpF method derived with the median value of $T_{opt}$ was combined with the SMAP_L3 surface SM product to estimate the subsurface SM. The subsurface SM was also compared to the SMAP_L4 root zone SM product (a widely used large-scale root zone SM product). Results indicated that the combination of the ExpF method with the SMAP_L3 surface SM product can significantly improve the estimation of profile SM in mountainous areas. Furthermore, the combination of SMAP_L3 and the ExpF method (with the median value of Topt) was applied to estimate the temporal and spatial distribution of profile SM in the study area.

2. For instance, I do not see any added value in the preliminary analysis of the soil moisture data. It is quite obvious that surface and subsurface soil moisture are linked or coupled. Remove this part or avoid stating that it is an outcome of the study.

Response:
We have deleted this part in the revised manuscript.

3. Second step in the manuscript is the intercomparison of different models. In this step, it seems that the use of ANN is made just applying a matlab tool without providing enough details about the approach adopted.

Response:

We have added the details of the ANN in the revised manuscript. (Line 153-165)

The ANN method is a data-driven method to predict subsurface SM from surface SM (Zhang et al., 2017a). If properly trained, ANN are able to describe nonlinear relationships between dynamics of SM at different depths (Kornelsen and Coulibaly, 2014). The commonly used feed-forward ANN (with one hidden layer and 10 neurons, Levenberg–Marquardt algorithm, Ford et al., 2014) was used in this study and the ANN modelling was carried out using MATLAB (neural network time series tool, R2017b, The MathWorks). The output of the ANN was calculated using:

$$y = f[W_2 g(W_1 X + b_1) + b_2]$$

where $y$ is the output, $f$ and $g$ is the activation function of the hidden layer and the input layer, respectively. $W_1$ and $W_2$ are the weights of input layer and hidden layer, respectively. $b1$ and $b2$ are the bias of input layer and hiden layer, respectively. The tangent sigmoid function was choiced as the activation function as it has the good performance in the hydrological studies (Yonaba et al., 2010). As suggested by Zhang et al. (2017a), 70% of data were randomly selected for training the ANN and the remaining 30% were used for validation. A separate ANN model was developed for every depth combination and every site.

4. The intercomparison may be influenced by the different approaches used for the calibration of the methods. In fact, authors states that 70% of the data was used for validation of ANN and CDF, but they do not provide such indication for the exponential filter. If they used the entire database for this last, this may affect the results.

Response:

The $T_{opt}$ parameter of the ExpF method reflects the characteristic length of the temporal dynamics of soil moisture. Earlier studies revealed that $T_{opt}$ is highly dependent on the sampling interval of soil moisture data (De Lange et al., 2008). In our study, we found that when using the random sampling with 70% training data as for the ANN method, $T_{opt}$ was not suitable for the remaining data. Since it was not possible to use the same training method for ExpF method as for ANN, we used the entire soil moisture time series to estimate $T_{opt}$, which was also the standard procedure in earlier studies (e.g. Wagner et al., 1999; Albergel et al., 2008; De Lange et al., 2008; Ford et al., 2014; Wang et al., 2017).

5. I personally do not understand the need to include a section of the cross-correlation analysis. It seems out of the scope of the manuscript. Moreover, no significant results are discussed herein. Please remove this section.

Response:

We have deleted the part of cross-correlation analysis in the revised manuscript.

6. Authors proposed some multilinear functions to describe relative value of T, which is fine, but it is not connected with anything else in the manuscript. It is another element somewhat independent from the main objective of the manuscript. Consider to

Response:

We have changed the manuscript. We deleted the regression equation for $T_{opt}$ that is not connected with the further analysis any more. Meanwhile, the evaluations of the other four methods for estimating $T_{opt}$ were kept, as the results indicated the usability of the median value of $T_{opt}$ for the ExpF method, which is important for data-scarce mountainous areas. What's more, the median value of $T_{opt}$ was used to derive the ExpF method to estimate the subsurface soil moisture from the SMAP_L3 surface product in the revised manuscript (Section 4.3.2). Thus, this part was connected to the further analysis in the revised manuscript.

7. In the last section, we start another new section where SMAP is used first in comparison with the observation revealing some limitation for higher values. Such statement should take into consideration the existing gap in the spatial resolution of the two measurements. Rough resolution tend to smooth out higher values. This is quite obvious.

Response:

We have noted the problem of scale mismatch between the in-situ observations and SMAP product. We have added the discussion about the introduced error from the scale mismatch in the revised manuscript. (Line 291-294)

Here, it is important to note that the SMAP_L3 product is provided at a 9 km × 9 km resolution while the in-situ measurements are point-based and soil moisture has a strong spatial variability in mountainous areas (Tian et al., 2019). Thus, part of the variability in Fig. 6 is due the disparity of spatial scales between the point-scale and the satellite footprint (Jin et al. 2017).

8. Finally, authors close with a comparison of exponential filter applied on SMAP. The regression are not used for this scope, other methods are not considered in this section, cross-correlation and spatial dynamics also neglected. I reached this point and I realized that authors are following a random walk of activities and I felt confused and disoriented.

Response:

We have made a drastic restructuring and reorganization in our revised manuscript. In the revised manuscript, we deleted the content related to the cross-correlation analysis and the regression analysis of Topt, which were not connected to the further analysis. After establishing that the median value of $T_{opt}$ can be used for the ExpF method, further calculation of subsurface soil moisture from SMAP_L3 surface soil moisture used the median value of $T_{opt}$ in the revised manuscript.

This manuscript requires a DRASTIC RESTRUCTURING and REORGANIZATION before being considered for publication. It will also benefit of a significant shortening of useless contents.

Response:
As stated above, we have revised the manuscript with a drastic restructuring and reorganization.

**Reference:**

Albergel C, Rüdiger C, Pellarin T et al. From near-surface to root-zone soil moisture using an exponential filter: an assessment of the method based on in-situ observations and model simulations. Hydrology and Earth System Sciences, 2008, 12(6): 1323-1337.

Ford T W, Harris E, Quiring S M. Estimating root zone soil moisture using near-surface observations from SMOS. Hydrology and Earth System Sciences, 2014, 18(1): 139-154.

Jin R, Li X, Liu S M. Understanding the Heterogeneity of Soil Moisture and Evapotranspiration Using Multiscale Observations From Satellites, Airborne Sensors, and a Ground-Based Observation Matrix. IEEE Geoscience and Remote Sensing Letters, 2017, 14(11): 2132-2136.

Kornelsen K C, Coulibaly P. Root‐zone soil moisture estimation using data‐driven methods. Water Resources Research, 2014, 50(4): 2946-2962.

Lange R d, Beck R, Giesen N v d, Friesen J, Wit A d, Wagner W. Scatterometer-Derived Soil Moisture Calibrated for Soil Texture With a One-Dimensional Water-Flow Model. IEEE transactions on Geoscience and remote sensing, 2008, 46(12): 4041-4049.

Tian J, Zhang B, He C, Han Z, Bogena H R, Huisman J A. Dynamic response patterns of profile soil moisture wetting events under different land covers in the Mountainous area of the Heihe River Watershed, Northwest China. Agricultural and Forest Meteorology, 2019, 271(15): 225-239.

Wagner W, Lemoine G, Rott H. A Method for Estimating Soil Moisture from ERS Scatterometer and Soil Data. Remote Sensing of Environment, 1999, 70(2): 191-207.

Wang T, Franz T E, You J, Shulski M D, Ray C. Evaluating controls of soil properties and climatic conditions on the use of an exponential filter for converting near surface to root zone soil moisture

contents. Journal of Hydrology, 2017, 548: 683-696.

Zhang N, Quiring S, Ochsner T, Ford T. Comparison of Three Methods for Vertical Extrapolation of Soil Moisture in Oklahoma. Vadose Zone Journal, 2017a, 16(10).

---

## Author Response (AR1)

Dear Editor,

We would like to thank both reviewers and you for your constructive comments and helpful suggestions that helped to improve the quality of our manuscript entitled "Estimation of subsurface soil moisture from surface soil moisture in cold mountainous areas".

We have taken into account all the reviewers' comments and thoroughly revised our manuscript with a major revision. In particular, we have now put a stronger emphasis on restructuring and reorganizing our manuscript logically to improve the scientific quality of our study and reanalyzing the results based on consistent statistics. Revisions following the comments of the reviewers are highlighted in yellow in the revised manuscript.

We are indebted to you and the reviewers for your taking significant amount of time and effort in handling our manuscript and providing detailed comments and suggestions for improving the manuscript. We look forward to your affirmative reply.

Sincerely,

Chansheng He, Ph.D. (on behalf of all co-authors)

Professor of Geography

Western Michigan University,

Kalamazoo, MI 49008, USA
* * *
**Responses to the comments from reviewers:**

The comments of reviewers are in **Bolded Arial font**, while our responses are indicated in Times New Roman font with blue color, and the New text passages are indicated in Times New Roman font with black color.

**Comments from Referee #1**

**The paper explores methods to estimate subsurface soil moisture from surface soil moisture based on an in-situ observations in cold mountainous areas since 2013. This variable is important for different scientific and applied topics. Authors explored the availability of three methods and applied the exponential filter method to the SMAP product. Research showed the improvement of profile soil moisture estimations in the mountainous. Many useful data, figures, and results were shown in the manuscript. I recommend a minor revision.**

**Response:** Thank you for your positive comments.

**General comments: The paper is well written, and the results are well presented. Bibliography very exhaustive. The analyzed dataset is interesting, and the results can be useful to improve the estimation of subsurface soil moisture and could be potentially useful for hydrological modelling. The results show that the combination of exponential filter method and satellite surface product can improve the estimation of profile soil moisture, and the availability of the area-generalized Topt in the cold mountainous areas. Related researches in high mountain ranges are limited around the world. Therefore, the presented results add new knowledge on those relevant hydrologic topics.**

**Response:** Thank you for your positive comments. We also think that our work can provide a useful reference for studies in high mountainous areas.

**1. Line 111, The half-hourly measurements were averaged to obtain daily SM values that will be used for the estimation of subsurface SM, which cover up the response of soil moisture to precipitation in a day if it's a rainstorm in where are a big soil porosity.**

**Response:** Yes. In this study, the soil moisture data was averaged from half-hourly scale to daily scale, and we neglect short-term effects of rainstorms on the soil moisture dynamics. The exponential filter (ExpF) method assumes that the water flux between two layers is proportional to the difference in soil moisture and that the temporal characteristics of soil moisture can be represented by one parameter (T, time characteristic length) (Albergel et al., 2008; Ceballos et al., 2005). This simplification of the ExpF method ignores the complex relationship between the surface and profile soil moisture during rainstorm events (e.g. Tian et al., 2019). Thus, the ExpF method is typically used at the daily time scale (e.g. Albergel et al., 2008; Ceballos et al., 2005; Ford et al., 2014; Wang et al., 2017).

**2. Figure 12, please explain this figure in detail about the temporal variation of soil moisture. It's obvious that SM increased in August. You can link the impact of climate change to moisture source and so on.**

**Response:** We have explained the figure in detail in the revised manuscript (Line 355-360).

"The temporal variation of profile SWI, surface SWI, and precipitation are shown in Fig. S10. Results showed that the temporal variation of profile SM corresponded well with the precipitation. It increased from May (with mean value of 0.27) to September (0.533), then decreased to October (0.304). Profile SWI$_{SMAP}$ was lower than surface SWI$_{SMAP}$ from May to August, while profile SWI$_{SMAP}$ was higher than surface SWI$_{SMAP}$ from September to October. This is attributed to the higher sensitivity of surface SM dynamics to precipitation and evapotranspiration (ET). During September and October, decreased precipitation and increased ET caused the faster decrease of surface SM compared to profile SM."

[Figure]

**Fig. S10** (a) the temporal variation of precipitation, SSWI and PSWI, and (b) the comparison (bar represents the mean value and error bar means the standard deviation) of the monthly SSWI and PSWI during the growing seasons of 2015-2017.

**Specific comments: Line number are related to the authors' line numbers.**

**3. Line 14, 'statistical' replace with 'multiple'.**

**Response:** We have changed it. (Line 14)

**4. Line 15, 'an' replace with 'its'.**

**Response:** We have changed it. (Line 14)

**5. Line 15, '15' and ＇ 25' replace with '10-20' and '20-30', respectively.**

**Response:** We have changed it. (Line 21)

**6. Line 25, 'with' replace with 'by'.**

**Response:** We have changed it. (Line 22)

**7. Line 26, please rewrite this sentence. I would prefer to 'the ExpF method was applied to estimate profile soil moisture using the satellite soil moisture product'.**

**Response:** We have changed it. (Line 23-24)

**8. Line 27, the first 'to' replace with 'with'.**

**Response:** We have changed it. (Line 25)

**9. Line 33, please insert 'as' before 'an'.**

**Response:** We have insert 'to be' before 'an'. (Line 31)

"Soil moisture (SM) is considered to be an essential climate variable"

**10. Line 36, 'included' replace with 'include'.**

**Response:** We have changed it. (Line 34)

**11. Line 41, 'provide' replace with 'provides'.**

**Response:** We have changed it. (Line 40)

**12. Line 48, 'from 0 to 60 cm depth' replace with '(from 0 to 60 cm depth) '.**

**Response:** As the cross-correlation analysis is not connected to the further analysis, we have deleted the part related to the cross-correlation analysis in the revised manuscript.

**13. Line 56, please delete 'that'.**

**Response:** We have changed it. (Line 50)

**14. Line 59, 'are' replace with 'is'.**

**Response:** We have changed it. (Line 54)

**15. Line 60, 'have' replace with 'has'.**

**Response:** We have changed it. (Line 54)

**16. Line 61, 'on' replace with 'about'.**

**Response:** We have rewritten the sentence. (Line 56-57)

"The exponential filter (ExpF) method belongs to the semi-empirical modeling approaches and relies on a two-layer SM balance equation (Wagner et al., 1999)."

**17. Line 73-74, please re-write this sentence 'In the absence of large‐scale networks of in situ SM observations in mountainous areas'.**

**Response:** We have changed it. (Line 67)

"In the absence of large-scale in-situ SM observations networks of mountainous areas"

**18. Line 84, delete 'which is'.**

**Response:** We have changed it. (Line 78)

**19. Line 106, please put the reference 'Zhang et al., 2017b' at the end of the sentence.**

**Response:** We have changed it. (Line 100)

**20. Line 107, ' (González-Zamora et al., 2016) ' replace with 'González-Zamora et al. (2016) '.**

**Response:** We have changed it. (Line 101)

**21. Line 109, 'data set' replace with 'dataset'.**

**Response:** We have changed it. (Line 103)

**22. Line 118, delete the 'and' after 'bulk density'.**

**Response:** We have changed it. (Line 112)

**23. Line 123, I think you use the SMAP products of 2015-2017 in your research, not only 2015-2016.**

**Response:** We have changed the '2015-2016' to '2015-2017'. (Line 117)

**24. Line 118, delete 'and' after 'bulk density'.**

**Response:** We have changed it. (Line 112)

**25. Line 155, 'tn-1' replace with 'tn-1'.**

 **Response:** We have changed it. (Line 146)

**26. Line 166, delete ',' before 'The ANN'.**

**Response:** We have changed it. (Line 157)

**27. Line 167, delete 'of' after 'training'.**

**Response:** We have changed it. (Line 164)

**28. Line 177, I think the equation (6) is incorrect, please correct it.**

**Response:** We have changed it. (Line 172-173)

$$\hat{\Delta} = K_0 + K_1 \cdot \theta_1 + K_2 \cdot \theta_1{}^2 + K_3 \cdot \theta_1{}^3 \qquad\qquad (6)$$

"Where $\hat{\Delta}$ is the predicted difference between surface and subsurface SM, and $K_i$ ($i$=0,1,2,3) are parameters."

**29. Line 190, please unify the 'lag time' and 'Lag time', I think it's better to use the term 'Lag time'.**

**Response:** As the response for comment 12, we have deleted the contents about the cross-correlation analysis. Thus, the Lag time was also deleted in the revised manuscript.

**30. Line 193, 'from 0-70 cm' replace with '(from 0-70 cm)'.**

**Response:** We have deleted this part.

**31. Line 204, add ','respectively at the end.**

**Response:** We have deleted this part.

**32. Line 211, I think you mean that 'no significant linear correlations' rather than 'no linear correlations'.**

**Response:** We have deleted this part.

**33. Line 216, delete 'have' after 'may'.**

**Response:** We have deleted this part.

**34. Line 250, 'season' replace with 'seasons'.**

**Response:** We have changed it. (Line 212)

**35. Line 270, insert 'ranging' before 'from'.**

**Response:** We have changed it. (Line 229)

**36. Line 280, please insert ' layer' before both the '3' and '4'.**

**Response:** We have changed it. (Line 238)

**37. Line 285, 'Topt' replace with 'Topt'.**

**Response**: Fig. 5 has been changed and merged with Fig. 4 into a new figure in the revised manuscript. (Fig. 4, Line 234)

**38. Line 300, 'suggest' replace with 'suggests'.**

**Response:** As suggested by referee 2, the correlation between ln-transformed LAI and precipitation is significant (Pearson's R=0.80, P<0.01). Furthermore, we tested the partial correlation analysis of the ln-transformed LAI, precipitation and Topt. The results showed that the relationships between ln-transformed LAI and $T_{opt}$ are nonsignificant under the control of precipitation. Meanwhile, the relationships between precipitation and $T_{opt}$ under the control of ln-transformed LAI are not valid for all layers. Thus, this section about the control factors of $T_{opt}$ is not convincing.

Furthermore, as the control factors and regression of Topt are not applied to the further estimation of subsurface soil moisture from the SMAP_L3 product, this part is not important for the manuscript.

Therefore, we have deleted the section about the control factors and regression of $T_{opt}$ in the revised manuscript.

**39. Line 310, I think it is negative correlations from Fig. 6.**

**Response:** As the response for comment 38, this part has been deleted in the revised manuscript.

**40. Line 356, 'Topt' replace with ' $T_{opt}$ '.**

**Response:** We have changed it. (Line 244)

**41. Line 380, insert ' were shown' before both 'in supplementary' and 'in Fig.11'.**

**Response:** We have changed it. (Line 282)

**42. Line 380, 'researches' replace with 'research'.**

**Response:** We have changed the sentence as following: (Line 284)

"The poor performance at scrubland sites is consistent with results presented by Zhang et al. (2017b) for this study region"

**43. Line 420, '$T_{opt}$' replace with '$T_{opt}$'.**

**Response:** We have changed it. (Line 337)

**44. Line 421, insert 'profile SWI' before 'estimation'.**

**Response:** This sentence has been deleted in the revised manuscript as we have rewritten the paragraph (Line 301-307)

"For the estimation of subsurface soil moisture from the SMAP_L3 surface product, the site-specific $T_{opt}$ was calculated based on the best match between SMAP estimations and in-situ observations in terms of NSE. The median values of $T_{opt}$

for the layers 2, 3, 4, 5 and profile are 7 days, 12 days, 22 days, 35 days and 10 days, respectively. The subsurface SWI estimated from the combination of SMAP surface soil moisture with the ExpF method (with the median values of $T_{opt}$) were compared with the in-situ observations. A comparison of the subsurface SWI time series for different layers at each station are provided in Fig. S3- S7. Fig.7 shows the scatter plot between measured and predicted SWI, and the performance metrics are summarized in Table 4."

**45. Line 430, 'season' replace with 'seasons'.**

**Response:** We have changed it. (Line 309)

**46. Line 444, 'soil profile moisture' replace with 'profile soil moisture'.**

**Response:** We have changed it. (Line 346)

**47. Line 447, 'SMAP-L4' replace with 'SMAP_L4'.**

**Response:** We have changed it. (Line 349)

**48. Line 485, insert 'The' before 'main findings'.**

**Response:** We have changed it. (Line 376)

**Comments from Referee #2**

**General comments:**

**This manuscript describes a wide-variety of approaches to estimate subsurface and profile soil moisture from surface soil moisture data in the Qilian Mountains of China. Most of the conclusions are well-supported, but the manuscript suffers from statistical inconsistencies. A consistent set of statistical measures should be maintained throughout the manuscript (NSE, RSR, and R). The only exception could be Fig. 10 where the aim is to compare the SMAP ubRMSE to the mission's accuracy requirement.**

**Response:** Thanks for your comments. We have maintained the consistency of the statistics throughout the manuscript, expect for the use of ubRMSE in the evaluation of SMAP product.

**Also, 30% of the data should be withheld for validation for all three methods, not only the ANN method.**

**Response:** Thanks for your comments. The $T_{opt}$ parameter of the ExpF method reflects the characteristic length of the temporal dynamics of soil moisture. Earlier studies revealed that $T_{opt}$ is highly dependent on the sampling interval of soil moisture data (De Lange et al., 2008). In our study, we found that when using the random sampling with 70% training data as for the ANN method, $T_{opt}$ was not suitable for the remaining data. Since it was not possible to use the same training method for ExpF method as for ANN, we used the entire soil moisture time series to estimate $T_{opt}$, which was also the standard procedure in earlier studies (e.g. Wagner et al., 1999; Albergel et al., 2008; De Lange et al., 2008; Ford et al., 2014; Wang et al., 2017).

**The performance of the SMAP-based ExpF method for estimating profile soil moisture (i.e., SWI) is overstated, and including the NSE and RSR statistics will likely provide a much more objective view.**

**Response:** Thanks for your comments. In the revised manuscript, the performance of SMAP-based ExpF method for estimating profile soil moisture has been reanalyzed using the metrics of NSE and RSR.

**The manuscript also suffers from poor organization in some places and is not well-written.**

**Response:** Thanks for your comments. We have deleted some content that is not important for the analysis and we have reorganized the text to make the revised manuscript easier to understand.

The revised manuscript is now divided into three parts. Firstly, we evaluated the different methods for estimating subsurface soil moisture (SM). The ExpF method was found to be the most suitable method for the further application in the study area.

Secondly, as the ExpF method has only one parameter ($T_{opt}$), it's importance to analyze the variation of $T_{opt}$. What's more, as the lack of large-scale SM in-situ observations in the high and cold mountainous areas, there is need to evaluate the alternative methods to estimate $T_{opt}$. And our results indicate that the median value of

$T_{opt}$ can be used for application of the ExpF method in the study area.

Finally, the ExpF method derived with the median value of $T_{opt}$ was combined with the SMAP_L3 surface SM product to estimate the subsurface SM. The subsurface SM was also compared to the SMAP_L4 root zone SM product (a widely used large-scale root zone SM product). Results indicated that the combination of the ExpF method with the SMAP_L3 surface SM product can significantly improve the estimation of profile SM in mountainous areas. Furthermore, the combination of the SMAP_L3 and ExpF method (with the median value of Topt) were applied to estimate the temporal and spatial distribution of profile SM in the study area.

We think that the revised manuscript is well organization and well written.

**Specific comments:**

**1. line 126. Clarify the meaning of "to consider the assumption of uniform vertical profiles of soil temperature and soil dielectric properties".**

**Response:** For passive microwave soil moisture retrieval soil temperature needs to be known. Considering a temperature gradient in soil and vegetation requires more complex retrieval methods. A gradient is expected during afternoon overpasses when insolation is able to heat the top canopy while shadows and transpiration cools lower vegetation parts. During morning overpasses a uniform temperature gradient can be assumed. However, the gradient in vegetation is mostly more significant than in the upper few soil centimeters. Therefore, we revised this sentence as follows: "to consider the assumption of uniform temperature profiles in the vegetation cover during morning overpasses."

"SMAP descending node observations acquired near 6:00 AM local solar time have been combined to global daily composites in order to reduce the impact of Faraday rotation and to consider the assumption of uniform temperature profiles in the vegetation cover during morning overpasses."

**2. Table 1. Results are being reported with too many significant figures. I doubt that the laboratory is able to measure sand and silt content with adequate precision to justify four significant figures. Reduce to a more appropriate level, perhaps three significant figures.**

**Response:** We have reduced the level to three significant figures in the revised manuscript:

Nevertheless, we think that our result about the sand and silt content are precision enough. Our results are measured by the Mastersizer-2000 (Malvern Inc.) The measurements of the grain-size (sand, silt and clay content) analyses were made with a Malvern Mastersizer 2000 laser grain-size analyser with a measurement range of 0.02-2000 μm at a 0.1 Φ resolution and an absolute error of <5%. The measurements are made following the standard procedures of the Mastersizer 2000 laser at the Key Laboratory of Western China's Environmental Systems (Ministry of Education), Lanzhou University. The measurement results are accurate

enough to be used for the scientific researches and has been used in many scientific papers (e.g. Sun et al., 2002, *Sedimentary Geology*; Sun et al., 2008, *Palaeogeography, Palaeoclimatology, Palaeoecology*; Zhang et al., 2016, *Quaternary Science Reviews*; Li et al., 2017, *Quaternary Science Reviews*; Guo et al., 2020, *Geophysical Research Letters*).

**Table 1**. Statistics of the soil physical characteristics at the 35 soil moisture stations: mean (standard deviation)

| Layer | Depth (cm) | Bulk Density (g/cm$^3$) | $K_S$ (cm/hour) | SOC (g/100g) | Sand (%) | Silt (%) | Clay (%) |
|---|---|---|---|---|---|---|---|
| Layer 1 | 0-10 | 1.13(0.28) | 3.87(4.11) | 4.35(4.11) | 26.6(11.9) | 66.2(10.9) | 7.2(1.6) |
| Layer 2 | 10-20 | 1.14(0.24) | 4.61(4.53) | 3.9(3.87) | 24.5(11.9) | 68.6(11.2) | 6.9(1.2) |
| Layer 3 | 20-30 | 1.18(0.32) | 4.78(6.22) | 3.63(3.54) | 27.0(15.2) | 66.5(14.3) | 6.5(1.4) |
| Layer 4 | 30-50 | 1.29(0.3) | 3.94(4.68) | 2.21(2.28) | 29.5(15.3) | 63.8(14.5) | 6.5(1.6) |
| Layer 5 | 50-70 | 1.34(0.3) | 1.85(2.35) | 2.34(2.47) | 26.9(17.1) | 66.5(15.9) | 6.7(1.9) |

Note: $K_S$ is the Saturated Hydraulic Conductivity; SOC is the Soil Organic Carbon.

**3. Section 3.4. Provide more explanation. Was a separate ANN model developed for every depth combination and every site?**

**Response:** Yes. In this study, a separate ANN model was developed for every depth of every site. We have added more details about the set-up of the ANN method in the revised manuscript (Section 3.3, Line 153-165).

"The ANN method is a data-driven method to predict subsurface SM from surface SM (Zhang et al., 2017a). If properly trained, ANN are able to describe nonlinear relationships between dynamics of SM at different depths (Kornelsen and Coulibaly, 2014). The commonly used feed-forward ANN (with one hidden layer and 10 neurons, Levenberg–Marquardt algorithm, Ford et al., 2014) was used in this study and the ANN modelling was carried out using MATLAB (neural network time series tool, R2017b, The MathWorks). The output of the ANN was calculated using:

$$y = f[W_2 g(W_1 X + b_1) + b_2] \tag{7}$$

where $y$ is the output, $f$ and $g$ are the activation functions of the hidden layer and the input layer, respectively, $W_1$ and $W_2$ are the weights of the input layer and the hidden layer, respectively, and $b1$ and $b2$ are the biases of the input layer and the hidden layer, respectively. The tangent sigmoid function was used as the activation function as it has shown good performance in hydrological studies (Yonaba et al., 2010). As suggested by Zhang et al. (2017a), 70% of data were randomly selected for training the ANN and the remaining 30% were used for validation. A separate ANN model was developed for every depth combination and every site."

**4. Equation 6 is the wrong equation.**

**Response:** We have corrected Equation 6 in the revised manuscript. (Line 173)

$$\hat{\Delta} = K_0 + K_1 \cdot \theta_1 + K_2 \cdot \theta_1^2 + K_3 \cdot \theta_1^3 \tag{9}$$

Where $\hat{\Delta}$ is the predicted difference between surface and subsurface SM, and $K_i$ (i=0,1,2,3) are parameters.

**5. line 196. not "persistent" but "consistent"**

Response: As suggested by referee 3, we have deleted this part.

**6. line 205-208. This should be moved to the methods section.**

Response: As suggested by referee 3, we have deleted this part.

**7. line 257. You have not provided any convincing evidence that "For most hydrological researches, the correct temporal variation of SM is more crucial than the exact value, suggesting that more emphasis should be given to R when selecting the most appropriate estimation method." You have presented three statistical measures to evaluate these methods (RSR, R, and NSE). For two out of the three statistical measures (RSR and NSE) the ANN method had the best performance. Therefore, you should include a clear statement that the results from the ANN method were statistically superior to those from the other two methods. You are still free to prefer the ExpF approach if it is simpler to apply than the ANN method. Just don't try to justify that choice on a statistical basis.**

Response: Thank you for your comment and suggestion. Yes, the ANN method is statistically superior to those from the other two methods. However, we prefer the ExpF approach as it is simpler to apply and more process-based than the ANN method, thus we can learn much from the ExpF method than the ANN method. We have deleted the statement "For most hydrological researches, the correct temporal variation of SM is more crucial than the exact value, suggesting that more emphasis should be given to R when selecting the most appropriate estimation method" in the revised manuscript. Furthermore, we have included the statement "The results suggested that for two out of the three statistical measures (RSR and NSE), the ANN method was statistically superior to those from the other two methods." in the revised manuscript. (Line 214-223)

"As expected, all metrics showed that the performance decreased with depth. The results indicate that for two out of the three statistical measures (i.e. RSR and NSE), the ANN method was statistically superior to the other two methods. Results also demonstrated that both ANN and ExpF method were able to provide accurate estimates of subsurface SM for layer 2, layer 3 and profile SM. Specifically, the ANN method resulted in the lowest estimation error, while the ExpF method was better able to capture SM dynamics. A similar finding was reported by Zhang et al. (2017a), who found that the ExpF method had a significantly higher correlation coefficient along with a higher mean bias compared to the ANN method. Furthermore, the ExpF method is a simpler approach as it only needs one parameter ($T_{opt}$), and can thus be easily applied in data-scarce mountainous areas, while the establishment of the ANN method is much more complicated. Besides, the ExpF method is a process-based method, while ANN is the machine learning method. Therefore, the ExpF method was used to estimate subsurface SM in the remainder of this study."

**8. Figure 7a is unnecessary and should be deleted. Your results show that "Year" does not have a significant effect, so the data should be presented including all years as done in Fig. 7b.**

**Response:** Thank you for your comment and suggestion. We have now merged Figure 5 into Figure 4 in the revised manuscript (Fig. 4).

[Figure]

**Fig. 4**. Variation of NSE with T of the exponential filter method at different layers of each stations during the growing season of 2014, 2015 and 2016. Y axis is the NSE value. Frequency distribution curve and the boxplots to show the distribution of $T_{opt}$ with depth for all stations.

**9. line 290-297. This should be moved to the methods section.**

**Response:** Thank you for your comment and suggestion. As suggested in Comment 10, the correlation among variables (e.g. Ln-transformed LAI and precipitation) are strong, and the results of partial correlation analysis among $T_{opt}$ and the variables indicated that the relationship among $T_{opt}$ and variables are not valid. Therefore, the relationships between $T_{opt}$ and variables are not applied in the further analysis. As a result, this part has been deleted in the revised manuscript.

**10. line 298-305. This section is not convincing. How strong is the correlation between ln-transformed LAI and precipitation? Perhaps the apparent relationship between Topt and LAI is a spurious result of the correlation between LAI and precipitation.**

**Response:** Thank you for your comment and suggestion. The correlation between ln-transformed LAI and precipitation are significant (Pearson's R=0.80, P<0.01). Furthermore, we tested the partial correlation analysis of the ln-transformed LAI, precipitation and $T_{opt}$. The results showed that the relationships between ln-transformed LAI and $T_{opt}$ are non-significant under the control of precipitation. Meanwhile, the relationships between precipitation and $T_{opt}$ under the control of ln-transformed LAI are not valid for all layers. Thus, this section on the control factors of $T_{opt}$ is not convincing and was removed.

Furthermore, as the control factors and regression of $T_{opt}$ are not applied any more to the further estimation of subsurface soil moisture from the SMAP_L3 product.

Thus, we have deleted the content on the control factors and regression of $T_{opt}$ in the revised manuscript.

**11. line 319-322. Move to methods.**

**Response:** We have moved it to the Methods section (Section 3.3).

**12. line 319-322. What steps were taken to prevent problems due to collinearity of the predictor variables?**

**Response:** Firstly, the regression equation was not used in the further analysis. Secondly, it is difficult to obtain independent variables for the regression equations of $T_{opt}$ in data-scarce mountainous areas. Thus, it's difficult to apply the regression equation for estimating $T_{opt}$ in the data-scarce mountainous areas. Therefore, we have deleted this part in the revised manuscript.

**13. Table 4. Is "ln ln (sand)" correct in the last row?**

**Response:** As stated in the reply for comment 10, this part has been deleted in the revised manuscript.

**14. Fig. 9. Present RSR instead of RMSE to be consistent with the rest of the manuscript.**

**Response:** We have changed the RMSE with RSR in the revised manuscript. (Fig. 5)

[Figure]

**Fig. 5.** The boxplot of NSE, Pearson's R, and RMSE for the $T_{opt}$ generated from different schemes. The different letters above box indicate the significant difference for different schemes.

**15. line 380. Not "persistent" but "consistent".**

Response: We have changed it. (Line 283)

**16. Section 4.4.1. You should note an important limitation of this analysis. There is a huge scale mismatch between the 9 km SMAP data and the in situ sensors which measure at a single point. This will likely degrade the agreement between the two data sets.**

Response: We agree that there is a large scale-disparity between our point measurements and the 9 km resolution of SMAP. Nevertheless, many studies have successfully compared point scale in-situ data with remote sensing information (e.g. Chen et al., 2017; Ford et al., 2014; González-Zamora et al., 2016; Paulik et al., 2014; Pablos et al., 2018; Ullah et al., 2018; Zhang et al., 2017). Some studies were able to downscale the SMAP soil moisture data to higher resolution with the aid of high resolution soil maps (Montzka et al. 2018). However, due to the harsh natural environment of the Qilian Mountain region, there is still lack of high-precision soil properties data (Li et al. 2017; Li et al., 2018; Tian et al. 2017; Zhao et al. 2018), precluding such downscaling treatments. Nevertheless, we used an extensive in-situ soil moisture station network with 36 soil moisture stations to cover the main vegetation, elevations and soil properties of the Qilian Mountain region. Therefore, we believe that our point scale measurement results are still providing sufficient spatial representativeness of the study area. Nevertheless, we have added a discussion on the potential influence of the scale mismatch between our datasets in the revised manuscript (Line 281-284).

"The relationship between the SMAP_L3 SM data product and the in-situ observations a 5 cm depth is presented in

Fig.6. Clearly, the part of the scatter in the relationship is due to the scale discrepancy between the satellite and the in-situ SM sensor data. Nevertheless, the statistical metrics still indicate a significant relationship between the SMAP_L3 SM data product and the in-situ observations a 5 cm depth."

**17. line 394-399. Move this to methods.**

**Response:** We have moved it to Data and Methods section (Section 3.1). (Line 122-124)

"SMAP_L3 surface soil moisture product was also used to estimate the subsurface soil moisture (Layer 2: 10-20 cm, Layer 3: 20-30 cm, Layer 4: 30-50 cm, Layer 5: 50-70 cm) and profile soil moisture (0-70 cm) during the growing seasons of 2015 and 2016 in the mountainous area."

**18. line 394-399. Why did you even bother all the effort to determine Topt from the in situ stations in the prior sections? Now you are not using those Topt values but instead finding new ones based on comparison of the SMAP data with the in situ data. This does not make sense in the flow of the manuscript.**

**Response:** On the one hand, the ExpF method only need one parameter ($T_{opt}$) to estimate the subsurface soil moisture. In order to use the exponential filter method in areas with limited or no soil moisture observations, especially in high and cold mountainous areas, it is necessary to understand the variation and the alternative estimations of $T_{opt}$. And this analysis showed that the median value of $T_{opt}$ is robust in the study area.

On the other hand, earlier studies revealed that $T_{opt}$ highly depended on the sampling interval of soil moisture data (De Lange et al., 2008). As data with the longer time interval typically have higher $T_{opt}$ values, $T_{opt}$ of in-situ data are mostly lower than satellite retrievals (Albergel et al. 2008; Ford et al., 2014; González-Zamora et al., 2016). In this study, the in-situ soil moisture observations are daily scale (the mean and standard deviation of the time interval for the 35 sites are 1 days and 0.3 days, respectively). However, as SM estimation from spaceborne sensors is still challenging for the high and cold mountainous regions, data gaps exist for the SMAP_L3 data in our study area (the mean and standard deviation of the time interval for the 35 sites are 2.3 days and 1.6 days, respectively). Thus, the time interval of the SMAP_L3 data is larger than the in-situ observations. Thus, the $T_{opt}$ of SMAP is different from the $T_{opt}$ of in-situ observations. Consequently, to get the best estimation of subsurface soil moisture at large scale, the subsurface SWI was estimated through combination of SMAP surface soil moisture with the ExpF method (with the site-specific $T_{opt}$ based on the best match between observations in terms of NSE).

**19. line 400-406. Include NSE and RSR measures here. They are crucial for quantifying the mismatch between the SMAP SWI and the observed SWI values as shown in Fig. 11.**

**Response:** We have included the NSE and RSR here. (Line 311-318)

"As expected, the estimation accuracy of subsurface SM decreased with depth. The ANOVA results showed that the

subsurface SM estimation accuracy for layer 2 (median value of RSR=0.92, R=0.69, NSE=0.18) and profile SM (RSR=0.92, R=0.65, NSE=0.14) were significantly higher than for layer 4 (RSR=1.12, R=0.31, NSE=-0.13) and layer 5 (RSR=1.17, R=0.34, NSE=-0.15) ($p<0.05$). The NSE values were positive for layer 2 and profile SM, while the NSE values for the other layers were negative. The negative MBE showed that subsurface SM was underestimated. The relationship between SMAP-derived and in-situ observed subsurface SM for layer 2 and profile SM was significant ($p<0.01$) at all but one station (D15). Thus, the SMAP surface product and ExpF method can be used to estimate the subsurface SM in the study area, especially for layer 2 (10-20 cm) and profile (0-70 cm) SM."

**20. line 425-428. This point should also have been made in Section 4.4.1.**

**Response:** We have added the error caused by the scale mismatch between point measurements and satellite footprints in Section 4.3.1 of the revised manuscript. (Line 281-284).

"The relationship between the SMAP_L3 SM data product and the in-situ observations a 5 cm depth is presented in Fig.6. Clearly, the part of the scatter in the relationship is due to the scale discrepancy between the satellite and the in-situ SM sensor data. Nevertheless, the statistical metrics still indicate a significant relationship between the SMAP_L3 SM data product and the in-situ observations a 5 cm depth."

**21. Tables 5 and 6. Replace RMSE with RSR. Add NSE.**

**Response:** We have replaced the RMSE with RSR and added the NSE in Tables 4 and 5. The analysis of the results was also changed according to the metrics of RSR, R and NSE in the revised manuscript. (Table 4 and Table 5, as the original Table 4 is the results of regression equation for $T_{opt}$, it has been deleted in the revised manuscript).

"**Table 4.** Statistics of the metrics (RSR, R, NSE) of the comparisons of SMAP estimated and observed SWI at different layers for the 35 stations during the growing seasons of 2015-2016.

| Layer | RSR | | R | | NSE | |
|---|---|---|---|---|---|---|
| | Mean±Std | Median | Mean±Std | Median | Mean±Std | Median |
| Layer 2 a | 1.21±1.27 | 0.90 | 0.64±0.24 | 0.73a | 0.04±0.49 | 0.19 |
| Layer 3 ab | 1.27±0.93 | 1.06 | 0.55±0.31 | 0.64ab | -0.05±0.46 | 0.05 |
| Layer 4 b | 1.53±1.49 | 1.10 | 0.43±0.38 | 0.52b | -0.22±0.53 | -0.08 |
| Layer 5 b | 2.03±3.48 | 1.16 | 0.41±0.39 | 0.55b | -0.27±0.61 | -0.14 |
| Profile a | 1.18±0.74 | 0.88 | 0.63±0.27 | 0.72a | 0.12±0.38 | 0.19 |

Note: the different letters after the layers indicate that the difference is significant at $p<0.05$ (Kruskal-Wallis ANOVA)

"**Table 5.** Statistics of the metrics (RSR, R, NSE) of the comparisons of estimated-observed profile SWI datasets, SMAP_L3-observed surface SWI datasets, SMAP_L3-observed profile SWI datasets, and SMAP_L4-observed profile SWI datasets for the 35 stations during the growing season of 2015-2016.

| Comparisons | RSR | | R | | NSE | |
|---|---|---|---|---|---|---|
| | Mean±std | Med | Mean±std | Med | Mean±std | Med |
| Estimated-observed PSWI | 0.86±1.00 | 0.68 | 0.88±0.11 | 0.9 | 0.56±0.32 | 0.64 |
| SMAP_L3-observed SSWI | 1.13±0.49 | 1.01 | 0.57±0.17 | 0.59 | -0.09±0.52 | -0.07 |
| SMAP_L3-observed PSWI | 1.18±0.74 | 0.88 | 0.63±0.27 | 0.72 | 0.12±0.38 | 0.19 |
| SMAP_L4-observed PSWI | 1.42±0.76 | 1.25 | 0.47±0.31 | 0.55 | -0.49±0.68 | -0.3 |

Note: e.g. Estimated-observed PSWI means the comparison of the estimated profile SWI and observed profile SWI. Med represents the median value.

Fig. S8 and Table 5 show that the SMAP-observed SWI comparisons had lower performance metrics for surface SWI (median value of RSR, R and NSE are 1.01, 0.59 and -0.07, respectively) than for profile SWI (median value of RSR, R and NSE are 0.88, 0.72 and 0.19, respectively). A possible reason for this is that the profile SWI was estimated based on the SMAP surface SWI and $T_{opt}$. The latter was determined by optimization using the maximum NSE, which improved the performance of profile SWI estimation. In addition, the performance metrics for SMAP–observed SWI comparisons for both surface and profile SWI were significantly (p<0.001) lower than those of estimated–observed profile SWI (median value of RSR, R and NSE are 0.68, 0.90 and 0.64, respectively). Thus, the major error in SMAP-based profile SWI estimates stems from the SMAP satellite product and is not derived from ExpF method, which is also supported by other researches (e.g. Ford et al., 2014; Pablos et al., 2018). Notably, the scale mismatch between point measurements and satellite footprints will introduce additional errors in the validation of the satellite estimated subsurface products (Jin et al. 2017)."

**22. line 436. Your results (Fig. 11) show that the performance of the SMAP profile SWI estimates is relatively poor. This is being partly obscured by the omission of the NSE and RSR statistics.**

**Response:** Thank you for your comments and suggests. We have added NSE and RSR in the revised manuscript. As shown in Tables 4 and 5, the SMAP estimated profile SWI has a median value of RSR=0.88, R=0.72 and NSE=0.19, respectively. For the estimation of profile soil moisture using the satellite surface soil moisture product and ExpF method in the previous studies: González-Zamora et al. (2016) found a mean R ranged from 0.6 to 0.8 (12 soil moisture stations of REMEDHUS network) for estimating the 0-50 cm soil moisture from the SMOS surface soil moisture. Ford et al. (2014) found a mean R=0.49 (0~0.71) and mean NSE=0.22 (-0.99~0.54) for the root zone soil moisture estimation from SMOS surface soil moisture at Nebraska station, USA. Thus, our results are comparable with the previous studies. Furthermore, although the NSE in this study is low, its performance is significantly (p<0.001) higher than the SMAP_L4 profile SWI (a widely-used root zone soil moisture product) in our study area.

**23. line 451-452. Here on the 22nd page of the manuscript a completely new data set is introduced. This is inappropriate. If this section is important to the manuscript, then take the time to justify it in the introduction and describe it in the methods.**

**Response:** We have deleted this part in the revised manuscript.

**24. line 465. Interpolated how? What evidence do you have that the interpolation is statistically valid? What is the associated uncertainty? This again should be justified in the introduction and described in the methods.**

**Response:** As suggested in comment 25, we can use the median value of $T_{opt}$ instead of using the site-specific $T_{opt}$ value. Thus, the median value of $T_{opt}$ ($T_{opt}$=10 days) was used to derive ExpF method in estimating the profile SWI in the study area in the revised manuscript. We have deleted the content related to the interpolation of $T_{opt}$ and the Fig. S14 (spatial distribution of $T_{opt}$).

**25. line 465. Also, why bother to spatially interpolate Topt? You have just argued that Topt defined in one region (Heihe) is valid in another region (Maqu).**

**Response:** As argued in Section 4.3.3, the median value of $T_{opt}$ can be used to derive the ExpF method for estimating profile soil moisture. For the estimation of profile SWI from SMAP surface product, we used the median value of $T_{opt}$ ($T_{opt}$=10 days) instead of the interpolated $T_{opt}$ to derive the ExpF method in the revised manuscript.

**26. line 495. The data in Fig. 11 show that the accuracy is relatively poor. Relying on the R value alone is clearly misleading in this case where there is a substantial bias. Including the NSE and RSR as suggested above will likely show that the performance is not very good.**

**Response:** We have included NSE and RSR in the analysis (Tables 4 and 5 in the revised manuscript). This is now mentioned in the revised manuscript accordingly. (Line 383-384)

"4) Subsurface SM derived from the SMAP_L3 surface SM product using the ExpF method showed less deviation from the in-situ observations compared to the SMAP_L4 root zone product for the study area."

**Comments from Referee #3**

This manuscript describes three approaches (ANN, Exponential filter, and CDF) to estimate subsurface soil moisture from surface soil moisture data in the Qilian Mountains of China. Authors identified the Exponential filter as the best model and applied this model in different ways throughout the manuscript. The topic is of great interest, but I think that the manuscript requires a significant restructuring in order to be considered acceptable for publication on HESS. My major concerns are:

**Response:** Thanks for your comments. We have made a significant restructuring in the revised manuscript.

**1. The organization of the manuscript and its presentation is not fluent. It seems that a series of tests and analysis have been listed one after the other without a logic.**

**Response**: Thanks for your comments. In the revised manuscript, we have deleted some contents that are not important for the analysis, and we made a drastic restructuring and reorganization to make the revised manuscript easier to understand.

The revised manuscript is now divided into three parts. Firstly, we evaluated the different methods for estimating subsurface soil moisture (SM). The ExpF method was found to be the most suitable method for further application in the study area.

Secondly, as the ExpF method has only one parameter ($T_{opt}$), it's importance to analyze the variation of $T_{opt}$. What's more, as the lack of large-scale SM in-situ observations in the high and cold mountainous areas, there is need to evaluate the alternative methods to estimate $T_{opt}$. And our results indicate that the median value of $T_{opt}$ can be used for application of the ExpF method in the study area.

Finally, the ExpF method derived with the median value of $T_{opt}$ was combined with the SMAP_L3 surface SM product to estimate the subsurface SM. The subsurface SM was also compared to the SMAP_L4 root zone SM product (a widely used large-scale root zone SM product). Results indicated that the combination of the ExpF method with the SMAP_L3 surface SM product can significantly improve the estimation of profile SM in mountainous areas. Furthermore, the combination of SMAP_L3 and the ExpF method (with the median value of Topt) was applied to estimate the temporal and spatial distribution of profile SM in the study area.

**2. For instance, I do not see any added value in the preliminary analysis of the soil moisture data. It is quite obvious that surface and subsurface soil moisture are linked or coupled. Remove this part or avoid stating that it is an outcome of the study.**

**Response:** We have deleted this part in the revised manuscript.

**3. Second step in the manuscript is the intercomparison of different models. In this step, it seems that the use of ANN is made just applying a matlab tool without providing enough details about the**

**approach adopted.**

**Response:** We have added the details of the ANN in the revised manuscript. (Line 154-165)

"The ANN method is a data-driven method to predict subsurface SM from surface SM (Zhang et al., 2017a). If properly trained, ANN are able to describe nonlinear relationships between dynamics of SM at different depths (Kornelsen and Coulibaly, 2014). The commonly used feed-forward ANN (with one hidden layer and 10 neurons, Levenberg–Marquardt algorithm, Ford et al., 2014) was used in this study and the ANN modelling was carried out using MATLAB (neural network time series tool, R2017b, The MathWorks). The output of the ANN was calculated using:

$$y = f[W_2 g(W_1 X + b_1) + b_2] \tag{7}$$

where y is the output, f and g is the activation function of the hidden layer and the input layer, respectively. $W_1$ and $W_2$ are the weights of input layer and hidden layer, respectively. b1 and b2 are the bias of input layer and hiden layer, respectively. The tangent sigmoid function was choiced as the activation function as it has the good performance in the hydrological studies (Yonaba et al., 2010). As suggested by Zhang et al. (2017a), 70% of data were randomly selected for training the ANN and the remaining 30% were used for validation. A separate ANN model was developed for every depth combination and every site."

**4. The intercomparison may be influenced by the different approaches used for the calibration of the methods. In fact, authors states that 70% of the data was used for validation of ANN and CDF, but they do not provide such indication for the exponential filter. If they used the entire database for this last, this may affect the results.**

**Response:** The $T_{opt}$ parameter of the ExpF method reflects the characteristic length of the temporal dynamics of soil moisture. Earlier studies revealed that $T_{opt}$ is highly dependent on the sampling interval of soil moisture data (De Lange et al., 2008). In our study, we found that when using the random sampling with 70% training data as for the ANN method, $T_{opt}$ was not suitable for the remaining data. Since it was not possible to use the same training method for ExpF method as for ANN, we used the entire soil moisture time series to estimate $T_{opt}$, which was also the standard procedure in earlier studies (e.g. Wagner et al., 1999; Albergel et al., 2008; De Lange et al., 2008; Ford et al., 2014; Wang et al., 2017).

**5. I personally do not understand the need to include a section of the cross-correlation analysis. It seems out of the scope of the manuscript. Moreover, no significant results are discussed herein. Please remove this section.**

**Response:** We have deleted the part of cross-correlation analysis in the revised manuscript.

**6. Authors proposed some multilinear functions to describe relative value of T, which is fine, but it is not connected with anything else in the manuscript. It is another element somewhat independent from the main objective of the manuscript. Consider to**

**Response:** We have changed the manuscript. We deleted the regression equation for $T_{opt}$ that is not connected with the further analysis any more. Meanwhile, the evaluations of the other four methods for estimating $T_{opt}$ were kept, as the results indicated the usability of the median value of $T_{opt}$ for the ExpF method, which is important for data-scarce mountainous areas. What's more, the median value of $T_{opt}$ was used to derive the ExpF method to estimate the subsurface soil moisture from the SMAP_L3 surface product in the revised manuscript (Section 4.3.2). Thus, this part was connected to the further analysis in the revised manuscript.

**7. In the last section, we start another new section where SMAP is used first in comparison with the observation revealing some limitation for higher values. Such statement should take into consideration the existing gap in the spatial resolution of the two measurements. Rough resolution tend to smooth out higher values. This is quite obvious.**

**Response:** We agree that there is a large scale-disparity between our point measurements and the 9 km resolution of SMAP. Nevertheless, many studies have successfully compared point scale in-situ data with remote sensing information (e.g. Chen et al., 2017; Ford et al., 2014; González-Zamora et al., 2016; Paulik et al., 2014; Pablos et al., 2018; Ullah et al., 2018; Zhang et al., 2017). Some studies were able to downscale the SMAP soil moisture data to higher resolution with the aid of high resolution soil maps (Montzka et al. 2018). However, due to the harsh natural environment of the Qilian Mountain region, there is still lack of high-precision soil properties data (Li et al. 2017; Li et al., 2018; Tian et al. 2017; Zhao et al. 2018), precluding such downscaling treatments. Nevertheless, we used an extensive in-situ soil moisture station network with 36 soil moisture stations to cover the main vegetation, elevations and soil properties of the Qilian Mountain region. Therefore, we believe that our point scale measurement results are still providing sufficient spatial representativeness of the study area.

Nevertheless, we have added the discussion about the introduced error from the scale mismatch in the revised manuscript. (Line 281-284)

"The relationship between the SMAP_L3 SM data product and the in-situ observations a 5 cm depth is presented in Fig.6. Clearly, the part of the scatter in the relationship is due to the scale discrepancy between the satellite and the in-situ SM sensor data. Nevertheless, the statistical metrics still indicate a significant relationship between the SMAP_L3 SM data product and the in-situ observations a 5 cm depth."

**8. Finally, authors close with a comparison of exponential filter applied on SMAP. The regression are not used for this scope, other methods are not considered in this section, cross-correlation and spatial dynamics also neglected. I reached this point and I realized that authors are following a random walk of activities and I felt confused and disoriented.**

**Response:** We have made a drastic restructuring and reorganization in our revised manuscript. In the revised manuscript, we deleted the content related to the cross-correlation analysis and the regression analysis of Topt, which were not connected to the further analysis. After establishing that the median value of $T_{opt}$ can be used for the

ExpF method, further calculation of subsurface soil moisture from SMAP_L3 surface soil moisture used the median value of $T_{opt}$ in the revised manuscript. As stated in detail in comment 1.

**This manuscript requires a DRASTIC RESTRUCTURING and REORGANIZATION before being considered for publication. It will also benefit of a significant shortening of useless contents.**

**Response:** As stated in detail in comment 1, we have revised the manuscript with a drastic restructuring and reorganization.

**Short Comments**

**1. The aim of this study is ambiguous. Is it comparison of different methods, improvement of methods, or evaluation of satellite products?**

**Response:** The aim of this study is to use multi-station in situ SM observations and remotely sensed SM data from the Qilian Mountains, a prime example of a high and cold mountainous area, to characterize the relationship between surface SM and deeper SM in order to obtain the spatial distribution of profile SM.

In the revised manuscript, we have deleted some contents that are not important for the analysis, and we made a drastic restructuring and reorganization to make the revised manuscript easier to understand.

The revised manuscript is now divided into three parts. Firstly, we evaluated the different methods for estimating subsurface soil moisture (SM). The ExpF method was found to be the most suitable method for further application in the study area.

Secondly, as the ExpF method has only one parameter ($T_{opt}$), it's importance to analyze the variation of $T_{opt}$. What's more, as the lack of large-scale SM in-situ observations in the high and cold mountainous areas, there is need to evaluate the alternative methods to estimate $T_{opt}$. And our results indicate that the median value of $T_{opt}$ can be used for application of the ExpF method in the study area.

Finally, the ExpF method derived with the median value of $T_{opt}$ was combined with the SMAP_L3 surface SM product to estimate the subsurface SM. The subsurface SM was also compared to the SMAP_L4 root zone SM product (a widely used large-scale root zone SM product). Results indicated that the combination of the ExpF method with the SMAP_L3 surface SM product can significantly improve the estimation of profile SM in mountainous areas. Furthermore, the combination of SMAP_L3 and the ExpF method (with the median value of Topt) was applied to estimate the temporal and spatial distribution of profile SM in the study area.

**2. Does the ExpF method with optimum Topt perform better than the ANN? If not, why does the author apply the ExpF method to expand SMAP?**

**Response:** Thank you for your comment and suggestion. The ANN method is statistically superior to those from the other two methods. However, we prefer the ExpF approach as it is simpler to apply and more process-based than the ANN method. in the revised manuscript. (Line 214-223)

"As expected, all metrics showed that the performance decreased with depth. The results indicate that for two out of the three statistical measures (i.e. RSR and NSE), the ANN method was statistically superior to the other two methods. Results also demonstrated that both ANN and ExpF method were able to provide accurate estimates of subsurface SM for layer 2, layer 3 and profile SM. Specifically, the ANN method resulted in the lowest estimation error, while the ExpF method was better able to capture SM dynamics. A similar finding was reported by Zhang et al. (2017a), who found that

the ExpF method had a significantly higher correlation coefficient along with a higher mean bias compared to the ANN method. Furthermore, the ExpF method is a simpler approach as it only needs one parameter ($T_{opt}$), and can thus be easily applied in data-scarce mountainous areas, while the establishment of the ANN method is much more complicated. Besides, the ExpF method is a process-based method, while ANN is the machine learning method. Therefore, the ExpF method was used to estimate subsurface SM in the remainder of this study."

**3. In the introduction, the author mentions there are four groups of methods, what are their advantages and disadvantages? Why did the author choose the three methods in this study?**

**Response:** In the introduction, we introduced five groups of methods, which are data assimilation of remote sensing data into land surface model (Han et al., 2013), physically-based methods (Manfreda et al., 2014), (semi-) empirical approaches (Albergel et al., 2008), data-driven methods (Kornelsen and Coulibaly, 2014; Zhang et al., 2017a), and statistical methods (Gao et al., 2019). Among them, the application of both data assimilation and physically based methods are limited due to the large amount of required input data, e.g. soil properties, which are often not available for data-scarce mountainous areas (Jin et al., 2015; Li et al., 2017; Dai et al., 2019). As we want to evaluate the methods that can be used in data-scarce mountainous areas, we exclude both data assimilation and physically based methods, and evaluated the other three methods in this study.

**4. In section 4.1, there are lag time between soil moisture data at different layers and at surface. How did the author consider the impacts of the lag time in applications of these methods?'**

**Response:** Thanks for your suggestion. We wanted to evaluate the coupling strength between surface soil moisture and subsurface soil moisture. However, there is no satisfying criterion to conclude whether the coupling strength is strong or not according to the results of the cross-correlation analysis. Thus, we have deleted the contents related to the cross-correlation analysis in the revised manuscript. So, we didn't consider the lag time in the applications of methods.

**5. The performance of the ANN method is significantly related to the training data. In this study, 70 % data was used as training data according to Zhang's study. However, Zhang's study focused on the US., is 70 % suitable for the high mountainous area? Moreover, even with a ratio of 70%, there are lots of data combinations, what's the principle to choose these data? Does the author compare the performance of the ANN method with different data combinations?**

**Response:** Firstly, for the ANN method, the sample number of training has no relation with the location, and 70% is usually selected as the number of the training samples (e.g. Kornelsen and Coulibaly, 2014; Zhang et al., 2017; ter Braak and Vrugt, 2008). The training with 70% of the data was also used in the estimation of soil moisture time series from passive microwave data using the ANN method in the Heihe River watershed (Lu et al., 2017).

Secondly, we used random sampling with uniform distribution in this study, which can best balance the

induced error by data sampling (Vrugt et al., 2011). Then, the combination with the best metric (minimum RMSE) was selected for the ANN method.

**6. In section 4.3.2, Topt is estimated by precipitation and clay ratio. However, the main advantage of the RBF method is its requirement of few data in introduction. Thus, the improvement in this study is meaningless. What's the insight of this improvement in other regions?**

**Response:** The correlation between ln-transformed LAI and precipitation is significant (Pearson's R=0.80, P<0.01). Furthermore, we tested the partial correlation analysis of the ln-transformed LAI, precipitation and Topt. The results showed that the relationships between ln-transformed LAI and $T_{opt}$ are nonsignificant under the control of precipitation. Meanwhile, the relationships between precipitation and $T_{opt}$ under the control of ln-transformed LAI are not valid for all layers. Thus, this section about the control factors of $T_{opt}$ is not convincing. Furthermore, as the control factors and regression of Topt are not applied to the further estimation of subsurface soil moisture from the SMAP_L3 product, this part is not important for the manuscript. Therefore, we have deleted the section on the control factors and regression of $T_{opt}$ in the revised manuscript.

We improved the estimation of profile soil moisture in data-scarce mountainous areas as follows. Our study evaluates four methods to estimate $T_{opt}$ in the data-scarce Qilian mountains, and the results show that the area-specific estimation of $T_{opt}$ (site-specific $T_{opt}$, area-generalized $T_{opt}$) has significantly higher performance than the widely-used $T_{opt}$ ($T_{Warger}$ and $T_{Qiu}$), which has been applied in cold mountainous areas (e.g. the utility of $T_{opt}$=20 days for profile SM estimation in east Asia in Muhammad et al. (2017)).

Furthermore, the results indicate that there is a non-significant difference between the performance of a site-specific $T_{opt}$ and an area-generalized $T_{opt}$. Thus, the area-specific Topt can be combined with ExpF method to estimate profile soil moisture with good performance. The reference $T_{opt}$ for the estimation of profile and subsurface soil moisture in the study area are provided in the manuscript, and provide a reference for future studies in similar areas.

Finally, we compared the estimation of profile soil moisture based on the combination of the SMAP_L3 surface product and the ExpF method (with a median value of $T_{opt}$ of SMAP) with a widely-used profile soil moisture product (SMAP_L4 root zone product). Results showed that our method can improve the profile soil moisture estimation significantly in our study area. Thus, based on the large-scale in-situ observations, we believe that our study improves the estimation of profile soil moisture in cold mountain areas, which will be useful for water resources management in inland river basins.

**7. The author evaluates both SMAP_L3 and SMAP_L4 products against in situ observations. The SMAP_L4 is the assimilation results of satellite data and model simulation. What's the impacts of their original biases of SMAP_L3 and SMAP_L4 respectively? What's the impacts of scale-mismatch between footprint scale of satellite products and point scale of in situ observations?**

**Response:** In this study, we have we partitioned the bias in the SMAP-based estimation of profile SWI ("SMAP-observed profile SWI", Fig. S8(c)) in bias associated with the ExpF method and bias due to SMAP original differences to get insight into the major source of error in SMAP-based estimates of profile SWI. Results showed that the major bias stems from the SMAP_L3 product.

The SMAP_L4 product is widely-used large-scale root zone soil moisture product. It is used as reference to test whether our method can improve profile soil moisture estimation. The results indicate that the ExpF method combined with the SMAP_L3 can significantly improve profile soil moisture estimation.

We have noted the problem of scale mismatch between the in-situ observations and the SMAP product. We have added a discussion about the introduced error from the scale mismatch in the revised manuscript (Line 281-284).

"The relationship between the SMAP_L3 SM data product and the in-situ observations a 5 cm depth is presented in Fig.6. Clearly, the part of the scatter in the relationship is due to the scale discrepancy between the satellite and the in-situ SM sensor data. Nevertheless, the statistical metrics still indicate a significant relationship between the SMAP_L3 SM data product and the in-situ observations a 5 cm depth."

[revised manuscript text omitted]

---

## Author Response (AR2)

Dear Editor,

We would like to thank both reviewers and you for your constructive comments and helpful suggestions that helped to improve the quality of our manuscript entitled "Estimation of subsurface soil moisture from surface soil moisture in cold mountainous areas".

We have taken the comments of the editor and reviewer#2 into account and revised our manuscript accordingly. In particular, we have now added an analysis of the methods comparison under the exact same condition (the first 70% of data is selected as training data, and the remaining 30% of data is used as validation data for the three methods). Moreover, we have thoroughly revised the entire manuscript to improve its readability. Revisions following the comments of the reviewers are highlighted in yellow in the revised manuscript.

We are indebted to you and the reviewers for your taking significant amount of time and effort in handling our manuscript and providing detailed comments and suggestions for improving the manuscript. We look forward to your affirmative reply.

Sincerely,

Chansheng He, Ph.D. (on behalf of all co-authors)

Professor of Geography

Western Michigan University,

Kalamazoo, MI 49008, USA
* * *
**Responses to the comments from reviewers:**

The comments of reviewers are in **Bolded Arial font**, while our responses are indicated in Times New Roman font with blue color, and the New text passages are indicated in Times New Roman font with black color.

**Comments from Editor**

**Dear Authors:**

**Your revised submission was evaluated by two out of the three previous reviewers. As you can see, while one reviewer gave a positive comment, the other was still quite critical. After an in-depth examination of both the last issues raised by Ref.#2 (of the revised paper), I agree with these additional concerns. Moreover, this part of the manuscript is indeed a bit hard to read and more flowing and especially clear sentences are required. Therefore, I invite you to provide adequate responses to the points raised by this Ref.#2.**

**As also raised by Ref.# 2 of the initial submission (who was different by the current #2), overall the manuscript still reads with some difficulty and I suggest you should improve the readability of the paper wherever possible.**

Response: Thanks for your comments. We have recalculated the intercomparison among the different methods and revised the manuscript according to the suggestion of reviewer#2. Please see the details in the response to reviewer#2.

This paper has been edited by a native English-speaker with a higher degree in a relevant discipline (www.GeoEditing.co.uk) to improve the readability of the paper (The picture below is the email with GeoEditing about polishing the English of the manuscript).

[Figure]

**Comments from Referee #2**

The manuscript have been improved significantly. I went thought the review and the rebuttal letter of the authors, I have to say that most of my comments have been addressed even if there is one issue that from my point of view has not been solved yet.

In my first review I requested more info about the dataset used for the calibration of the methods. For sake of clarity, I report in the following the comment:

"The intercomparison may be influenced by the different approaches used for the calibration of the methods. In fact, authors states that 70% of the data was used for validation of ANN and CDF, but they do not provide such indication for the exponential filter. If they used the entire database for this last, this may affect the results."

Based on the reply the authors clarified that the only for ANN a portion of the dataset have been used, while for the exponential filter the entire dataset was used. This means that the comparison between the two method is not made under the same conditions. This is a critical element that do not allow to discriminate among the models.

I do not agree with the author when they state that the exponential filter requires the entire dataset for the its application. It can be easily calibrated on a portion of the time series (the first 70% or a continuous window that covers such length of data). Therefore, I strongly encourage to repeat the analysis using continuous subset of soil moisture for both methods in order to obtain comparable results.

**Response:** Thanks for your comments. To make sure the comparison among the three methods is made under the same conditions, the first 70% of the data was selected as the training data, and the remaining 30% of the data was used as validation data for the three methods. In the previous study, the training data was selected using the random sampling with the best metric (minimum RMSE). Thus, the divide of the first 70% of data into training data will influence the performance of the three methods.

Our results show that ANN performed better than ExpF for the individual layers (layer 1 to 5) in terms of both NSE and RSR (Table S1 and Fig. S2). However, the ExpF method performed better than the ANN method in estimating soil moisture for the entire soil profile. Additionally, the comparison of the performances between the ExpF and ANN methods were nonsignificant ($p>0.05$) for all the layers. However, metrics showed that the ExpF method has significantly higher R value than the ANN method for all layers ($p<0.05$). Comparison of metrics illustrated that the CDF matching method has the lowest performance among the three methods. In conclusion, the metrics indicated that the ANN has the best performance in terms of NSE and RSR, the ExpF method has the best performance in terms of R value. Thus, the conclusion in the nee setup is similar to the conclusions of the previous setup. 
[revised manuscript text omitted]

[Figure]

**Fig. S10.** (a) the temporal variation of precipitation, SSWI and PSWI, and (b) the comparison (bar represents the mean value and error bar means the standard deviation) of the monthly SSWI and PSWI during the growing seasons of 2015-2017.